# On the Uncomputability of Partition Functions in Energy-Based Sequence Models

**Chu-Cheng Lin & Arya D. McCarthy**
Center for Language and Speech Processing, Johns Hopkins University

## Abstract

In this paper, we argue that energy-based sequence models backed by expressive parametric families can result in uncomputable and inapproximable partition functions. Among other things, this makes model selection — and therefore learning model parameters — not only difficult, but generally *undecidable*. The reason is that there are no good deterministic or randomized estimators of partition functions. Specifically, we exhibit a pathological example where under common assumptions, *no* useful importance sampling estimators of the partition function can guarantee to have variance bounded below a rational number. As alternatives, we consider sequence model families whose partition functions are computable (if they exist), but at the cost of reduced expressiveness. Our theoretical results suggest that statistical procedures with asymptotic guarantees and sheer (but finite) amounts of compute are not the only things that make sequence modeling work; computability concerns must not be neglected as we consider more expressive model parametrizations.

## 1 Introduction

Modeling discrete sequences is central to natural language processing and bioinformatics. Many common parametric sequence models $\tilde{p}$ are (or can be cast as) *energy-based* (LeCun et al., 2006): they yield a weight $\tilde{p}(\boldsymbol{x})$ for any given string $\boldsymbol{x}$. Although energy-based models (EBMs) of sequences need not represent probability distributions, they have been proposed as such to counter the inexpressivity of autoregressive sequence models (Bakhtin et al., 2021; Lin et al., 2021, §4.1). EBMs with finite **partition functions** $Z_{\tilde{p}} \triangleq \sum_{\boldsymbol{x}} \tilde{p}(\boldsymbol{x}) \in \mathbb{R}_{>0}$ define distributions $p(\boldsymbol{x}) \triangleq \tilde{p}(\boldsymbol{x})/Z_{\tilde{p}}$ over strings, which are often queried under the principle of probabilistic inference (Ghahramani, 2015).[1]

Energy-based sequence models are often parametrized as neural models. Each parameter vector $\boldsymbol{\theta}$ in some parametric neural network family $\boldsymbol{\Theta} \subseteq \mathbb{R}^d$ identifies a parametric model $\tilde{p}_{\boldsymbol{\theta}}$, which then defines a parametric distribution over strings $p_{\boldsymbol{\theta}}$, assuming $Z_{\tilde{p}_{\boldsymbol{\theta}}}$ exists (Chen et al., 2018). Contemporary neural networks have been shown to be very powerful. In particular, some popular parametric neural sequence model families, such as RNNs (Siegelmann & Sontag, 1995) and Transformers (Bhattamishra et al., 2020; Pérez et al., 2021), have been formally shown to recognize recursively enumerable languages: given *any* Turing machine $M$, there is a parameter vector $\boldsymbol{\theta} \in \boldsymbol{\Theta}$ such that the parametrized sequence model $N_{\boldsymbol{\theta}}$ recognizes the same language as $M$ does. In other words, these sequence model families are **Turing-complete**.

It is therefore intuitive that energy-based sequence models, backed by such powerful neural networks, form expressive families of string distributions (§3.1): for *any* decision algorithm $M$, there is a parameter vector $\boldsymbol{\theta}_M$ in a Turing-complete parametric family $\boldsymbol{\Theta}$, such that $p_{\boldsymbol{\theta}_M}$ exists, and $p_{\boldsymbol{\theta}_M}(\boldsymbol{x})$ is high if and only if $M$ accepts $\boldsymbol{x}$. It would seem assuring to work with such expressive family of sequence distributions, $\boldsymbol{\Theta}$: assuming the true string probabilities indeed can be computed in polytime, $\boldsymbol{\Theta}$ is well-specified. Moreover, one may assume that given we can sample from $p_{\boldsymbol{\theta}_M}$, we would be able to use consistent estimators to find $\boldsymbol{\theta}' \in \boldsymbol{\Theta}$, where $\boldsymbol{\theta}' \approx \boldsymbol{\theta}_M$.

Unfortunately, we find that with such an expressive distribution family $\boldsymbol{\Theta}$, whether the identifiability assumption holds — required for most consistent estimators — itself is **undecidable** (Turing, 1937).

---

[1]Many popular energy-based sequence models compute *normalized* probabilities directly (*i.e.*, $\tilde{p}(\boldsymbol{x}) = p(\boldsymbol{x})$) (Jelinek, 1980; Katz, 1987; Bengio et al., 2003; Brown et al., 2020, *inter alia*). This makes both training and querying with string prefixes much easier, at the cost of expressiveness (Lin et al., 2021).

Moreover, model selection on *any* held-out data is also undecidable. Even worse, we show that there exists a vector $\theta' \in \Theta$, such that as long as a parametric family $\Theta' \subseteq \Theta$ contains $\theta'$, model selection will not be possible for $\Theta'$ either — even when $\Theta'$ itself is not necessarily expressive (*e.g.*, $\Theta'$ can be fixed-size Transformer EBMs, which cannot parametrize all EBMs that require more parameters). We construct one such 'pathological' distribution $p_{\theta'}$ as a Transformer EBM.

These negative results stem from the uncomputability and inapproximability of $Z$. A main technical contribution of this paper is that there is no algorithm (either deterministic or randomized) that can approximate $Z$ well. An immediate consequence is that sampling-based statistical procedures are not useful in this scenario, either, as long as they terminate in finite time. However, we will see that for less expressive model families, such uncomputability issues do not arise. Our negative results are summarized in Table A.1. To ensure that model selection is still possible (such that we can compare different parameter vectors with confidence),[2] we have no choice but to resort to less expressive string distributions.

The paper is structured as follows. In §2 we review definitions and known results of weighted languages, sequence model families, and formalize weighted Turing machines, EC-complete parametric families, and computable estimators. In §§3–5 we describe our main technical results: there exist pathological EBM sequence models that have uncomputable partition functions, which cannot be approximated well under randomized estimators, and do not have asymptotic estimators that have any good guarantees. In §6 we that argue our negative results make model selection impossible for expressive model families, and discuss why common estimation methods fail. Finally, we discuss three 'palliative' parametrization choices in §7, which all guarantee computable partition functions, at the cost of expressiveness.

## 2 BACKGROUND

### 2.1 ENERGY-BASED SEQUENCE MODELS

Energy-based models (EBMs) of sequences (LeCun et al., 2006; Rosenfeld et al., 2001; Sandbank, 2008; Huang et al., 2018; Bakhtin et al., 2021) are a family of discrete sequence models. Given a string $x \in V^*$ over a finite vocabulary $V$, an EBM $\tilde{p}$ computes string **weight** $\tilde{p}(x)$, but not the (normalized) string **probability** $p(x) = \tilde{p}(x)/\sum_x \tilde{p}(x)$. In this work, we focus on EBMs whose weight is efficiently computable, *i.e.*, polytime in the length of $x$. Previous work (Bakhtin et al., 2021; Lin et al., 2021) showed that EBMs define expressive distributions; but this requires normalization. While EBMs are often intractable to normalize (*e.g.*, the Ising model), the infinite domain of finite strings opens the door to *uncomputable* probabilities, which invite the impossibility of model selection and comparison.

### 2.2 WEIGHTED LANGUAGES

EBMs give weights to strings. Here we formally characterize these weighted strings as a *weighted language*. An unweighted language $L \subseteq V^*$ is a set of finite strings $x$ over a finite vocabulary $V$. A **weighted language** $\tilde{p}$ is a function $\tilde{p} : V^* \rightarrow \mathbb{R}_{\geq 0}$.[3] In this work, we discuss Boolean languages, such that $V = \mathbb{B} \triangleq \{0, 1\}$. We also focus on weighted languages where distributions over strings exist. Following Lin et al. (2021), we say such weighted languages $\tilde{p}$ are **normalizable**: $Z_{\tilde{p}} \triangleq \sum_{x \in \mathbb{B}^*} \tilde{p}(x) \in \mathbb{R}_{>0}$. $Z_{\tilde{p}}$ is also called the **partition function** of $\tilde{p}$.[4] We can then normalize $\tilde{p}$ into a **distribution** $p$ over $\mathbb{B}^*$ such that $p(x) = \tilde{p}(x)/Z_{\tilde{p}}$, and thereby $\sum_{x \in \mathbb{B}^*} p(x) = 1$.

The **efficiently computable (weighted) languages** (EC; Lin et al., 2021) are those weighted languages $\tilde{p}$, where a string's weight (which must be non-negative) is a polytime function of the string. Weighted languages defined by most EBM sequence models fall into this class, since they score every string in finite time (and usually in polytime); and the scores ultimately are computed by some algorithm.

Intuitively, string weights of an EC language can be obtained as a side product from a **weighted Turing machine** that recognizes $x$ (in polytime). However, Lin et al. (2021) did not precisely describe how a Turing machine maps input string $x$ to its (rational) weight $\tilde{p}(x)$ in finite time. In this work, such a construction (out of many possible ones) is necessary, as we need to show that this string weighting can be

---

[2]Of course, model selection (and computing $Z_{\tilde{p}}$) are not always needed — for example, simply deciding whether $\tilde{p}(x_1) > \tilde{p}(x_2)$ for two given strings $x_{\{1,2\}}$. In such case the uncomputability issues we discuss are not a concern.

[3]With a slight abuse of notation, we also define support$(\tilde{p}) \triangleq \{x : \tilde{p}(x) > 0\}$.

[4]We use the convention that $Z_{\tilde{p}}$ is the partition function of $\tilde{p}$, and $Z_{\tilde{q}}$ of $\tilde{q}$, etc. The subscript is omitted in unambiguous cases.

done by parametric sequence model families (see Appendix B).

## 2.3 Locally normalized distributions

A popular subclass of weighted languages is **locally normalized weighted languages** (LN), where conditional local distributions given prefix $\hat{x}$: $p(\cdot \mid \hat{x})$ can be computed in finite time. Since they automatically define a distribution over strings, we use the term **locally normalized distributions** interchangeably. If $p(\cdot \mid \hat{x})$ can be computed in polynomial time, we call such distributions **efficiently locally normalized distributions** (ELN) — this is the weighted language class of most autoregressive models, just as its superset EC is the weighted language class of most energy-based sequence models.

Locally normalized distributions over sequences have $\tilde{p}(x) = p(x)$, and $Z = \sum_{x \in \mathbb{B}^*} \tilde{p}(x) = 1$. They are **consistent**: the probability that a string is infinitely long under such distributions is zero. Equivalently, given any $\epsilon > 0$, we can approximate $Z$ with a finite sum of string weights:

**Proposition 1** (Consistency of LN distributions (Booth & Thompson, 1973; Chen et al., 2018)). *Let* $p \in \text{LN}$ *be a locally normalized distribution over strings. All strings of a given length or longer have their probabilities bounded. That is, for all positive real numbers $\epsilon$, there is a length n at which all strings $x$ of length at least n have $p(x) < \epsilon$.*

The consistency property of locally normalized distributions implies that they have an exact sampling procedure that almost surely terminates in finite time. Therefore, they (and in particular ELN distributions) are an attractive choice when we need to sample from the distribution they define, *e.g.*, in sampling-based parameter estimation procedures.

## 2.4 EC-complete parametric families

We have introduced energy-based sequence models (§2.1) and their characterization as weighted languages (§2.2). However, we usually do not work with weighted Turing machines directly in machine learning. Here the most common models of computation are (neural) **sequence model families**, such as RNNs and Transformers. While these computational models can appear quite dissimilar to state-machine-based models of computation (*e.g.*, Turing machines), they have been shown to possess the same computation power (Siegelmann & Sontag, 1995; Pérez et al., 2021). That is, they are *Turing-complete*.

Just as we extend the definition of Turing machines to weighted Turing machines, we likewise formalize *weighted* sequence model families. We thus introduce EC**-complete parametric families** as a sequence model counterpart of the weighted language class EC. At a high level, a parametric family $\Theta$ is EC-complete if given any $\tilde{p} \in \text{EC}$ (as a description of a weighted Turing machine), we can construct a parameter vector $\theta \in \Theta \subseteq \mathbb{Q}^*$ which defines $\tilde{p}$'s corresponding sequence model. The model produces an output embedding which we then decide in polytime to be either a 'halting embedding' or not. If it is, we can then extract the weight $\tilde{p}(x)$. (A rigorous exposition is in Appendix C.)

We show that with a modification to positional embeddings, the family of arbitrary precision, hard attention Transformers defined in Pérez et al. (2021) is EC-complete:[5]

**Theorem 1.** *The class of one-encoder-layer, four-decoder-layer Transformer networks with positional encodings $(n, 1/n, 1/n^2, 2^n)$ is EC-complete.*

*Proof sketch.* We extend the construction from Pérez et al. (2021), adding an additional layer to accumulate the string weight over time steps. □

## 2.5 Estimators

A main result of this work is that partition functions in an EC-complete family are uncomputable. Moreover, randomness does not help estimation; and correct asymptotic estimators are not useful. We define these estimators here in order to discuss the power of different estimators concretely.

Let $\Theta$ be a parametric family. The function $f : \Theta \to \mathbb{Q}$ is an **exact estimator** if there exists a weighted deterministic Turing machine that takes $\theta$ as input and, upon halting, outputs $f(\theta) \in \mathbb{Q}$ in finite time.

---

[5]All proofs omitted from the main text are included in Appendix D. Proof sketches for major theorems are in the main text.

Many estimation procedures are *anytime algorithms*: they do not have a predetermined runtime, and they can be stopped at any time before completion to produce a valid output. Output quality of an anytime algorithm may improve with increased runtime. Moreover, many of these algorithms have asymptotic guarantees, in that their outputs are optimal in some sense (*e.g.*, consistency) in the limit. We capture these algorithms with **asymptotic estimators**: a function $f(\boldsymbol{\theta})$ is an asymptotic estimator if there exists a weighted Turing machine that takes both $\boldsymbol{\theta}$ and an index $i$ as input, then outputs a value $f_{\tilde{p},i} \in \mathbb{Q}$ in finite time, such that the outputs converge toward $f(\boldsymbol{\theta})$ as $i$ increases toward infinity. (An example is $\hat{Z}_{\text{asym}}$ introduced at §5.)

We now extend both exact and asymptotic estimators to the stochastic case, where we compute the estimates using randomized algorithms instead of deterministic ones. As is conventional for randomized algorithms, we assume the use of probabilistic Turing machines. These have access to an infinitely long **random tape**. The tape describes an infinite binary sequence from tossing a fair two-sided coin infinitely many times. We call the random tape distribution $p_\tau$.[6] We define a **randomized exact estimator f** as a weighted Turing machine with two input tapes — the random tape $\tau \in \mathbb{B}^{\mathbb{N}}$, and an input tape that has $\boldsymbol{\theta}$ — and outputs $f(\boldsymbol{\theta}, \tau)$ in finite time. Likewise, we say $\mathbf{f}_{\boldsymbol{\theta},i}$ is a **randomized asymptotic estimator** if there exists a function $f(\boldsymbol{\theta}) \in \mathbb{R}$ and a weighted Turing machine that takes $(\boldsymbol{\theta}, i), \tau$ on two input tapes, so that for all random Boolean tapes $\tau \in \mathbb{B}^{\mathbb{N}}$, we converge with $\lim_{i\to\infty} f_{\boldsymbol{\theta},i,\tau} = f(\boldsymbol{\theta})$. Many Monte Carlo estimators can be seen as randomized asymptotic estimators, including rejection sampling and importance sampling estimators.

# 3 EXPRESSIVENESS AND UNCOMPUTABILITY: PATHOLOGICAL EBMs

## 3.1 EXPRESSIVE SEQUENCE DISTRIBUTIONS

To illustrate the uncomputability issues of energy-based models, we construct families of string distributions that are **expressive**: they require the full power of an EC-complete sequence model family.

We define $\mathcal{G}_k = \{\tilde{p}_{M,k} : k \in \mathbb{Q}_{>0}\}$ to be a set of weighted languages, where $\tilde{p}_{M,k}$ is parametrized by deterministic Turing machine $M$ that takes an empty string as input ('input-free'). Let $L_M \subseteq \mathbb{B}^*$ be a prefix-free Boolean language of computation sequences of a Turing machine — that is, encodings of the trace of the Turing machine $M$'s operations. We define

$$\tilde{p}_{M,k}(\boldsymbol{x}) = \begin{cases} 1/3^{|\boldsymbol{x}|+1} + k & \text{if } \boldsymbol{x} \in L_M, \text{ and } \boldsymbol{x} \text{ encodes a valid accepting} \\ & \text{trace of } M. \\ 1/3^{|\boldsymbol{x}|+1} & \text{otherwise.} \end{cases} \tag{1}$$

The weight of any string $\boldsymbol{x}$, where $|\boldsymbol{x}| = n$, under $\tilde{p}_{M,k}$ can be computed in time $O\big(\text{poly}(n)\big)$, by verifying whether $\boldsymbol{x}$ is an accepting execution trace of $M$, from the initial state to an halting state. That is, $\tilde{p} \in \text{EC}$ (§2.2). We also know that for any (deterministic) machine $M$, the language's partition function $Z_{\tilde{p}_{M,k}}$ exists, and it must equal either 1 or $1 + k$, since there is either one halting trace ($Z_{\tilde{p}_{M,k}} = 1 + k$), or none ($Z_{\tilde{p}_{M,k}} = 1$). Therefore, each $\tilde{p} \in \mathcal{G}_k$ defines a string distribution.[7]

Since for all $k \in \mathbb{Q}_{>0}$, $\mathcal{G}_k \subset \text{EC}$, all weighted languages $\tilde{p}_{M,k}$ have an equivalent parameter vector $\boldsymbol{\theta}_{M,k}$ in any EC-complete family $\boldsymbol{\Theta}$. Also, since each $\mathcal{G}_k$ is a bijection between the set of all input-free Turing machines and a subset of $\boldsymbol{\Theta}$, it follows that there is no exact estimator (§2.5) of the partition function of any EC-complete family (*e.g.*, Transformer EBMs (Theorem 1)), by a reduction from HALT:[8]

**Theorem 2.** *Let $\boldsymbol{\Theta}$ be a parametric EC-complete sequence model family. And let $\boldsymbol{\Theta}_k \subset \boldsymbol{\Theta}$ be bijective with $\mathcal{G}_k$. There is no $k \in \mathbb{Q}_{>0}$ for which there exists an exact estimator $\hat{Z}_k$ that takes as input $\boldsymbol{\theta} \in \boldsymbol{\Theta}_k$ as input, and computes $\hat{Z}_k(\boldsymbol{\theta}) = Z_{\tilde{p}_{\boldsymbol{\theta}}}$ in finite time.*

*Proof sketch.* $\boldsymbol{\Theta}_k$ contains parametric vectors for all input-free Turing machines $M$. For any of these

---

[6]Formally speaking, we define a probability space $(\Omega, \mathcal{A}, \mathbb{P})$ where $\Omega = \mathbb{B}^{\mathbb{N}}$ is our sample space, $\mathcal{A} = \{A_{\boldsymbol{b}} : A_{\boldsymbol{b}}$ is the set of all sequences $\in \Omega$ that have prefix $\boldsymbol{b} \in \mathbb{B}^*\}$ is our event space, and $\mathbb{P}(A) = 2^{-n}$ where $n$ is the length of the longest shared prefix of $A$, is our probability function (Stroock, 2014).

[7]Our construction of expressive sequence distributions is inspired by a weighted language construction in Lin et al. (2021). See Appendix E for further discussion.

[8]Speaking loosely, HALT is the task of recognizing (deciding) whether a given program on an ideal computer will properly terminate in finite time or not (Sipser, 2013; Arora & Barak, 2006).

weighted languages, knowing the exact value of $Z_{\tilde{p}_{M,k}} \in \{1, k+1\}$ is enough to decide whether $M$ halts ($M$ halts iff $Z_{\tilde{p}_{M,j}} = k+1$). As HALT is undecidable for input-free Turing machines, $\hat{Z}_k$ cannot exist. $\square$

## 3.2 AN EBM WHOSE PARTITION FUNCTION IS UNCOMPUTABLE

Theorem 2 states there is no estimator that 'works' for a subset of parameter vectors. While every $\mathcal{G}_k$ is much smaller than its superset EC, $\mathcal{G}_k$ is still (countably) infinite. Here under the assumption that ZFC is consistent, we construct *one* weighted language $\tilde{b} \in \mathcal{G}_1$ (for simplicity; it holds for arbitrary $k$), where $Z_{\tilde{b}}$ is uncomputable:

**Theorem 3.** *Assuming ZFC axioms and that assuming they is consistent, there exists an* EC *weighted language $\tilde{b} \in \mathcal{G}_1$ such that (a) $Z_{\tilde{b}} = c \in \{1, 2\}$ exists, but (b) there is no proof of $Z_{\tilde{b}} = c$.*

*Proof sketch.* We construct $\tilde{b}$ such that $Z_{\tilde{b}} = 2$ iff ZFC is inconsistent. If there were a proof $Z_{\tilde{b}} = 1$ then we proved ZFC is consistent, which is impossible under Gödel's 2nd incompleteness theorem (Jech, 1994). If there were a proof $Z_{\tilde{b}} = 2$, then we proved ZFC is not consistent, which violated our assumption. $\square$

The existence of $\tilde{b}$ suggests that if there is an algorithm that approximates $Z_{\tilde{p}}$, *and* produces a witness of its approximation quality, then this algorithm will not work on *any* set of parameter vectors that can parametrize $\tilde{b}$ — even when this algorithm may work for some subsets of $\Theta$. This is useful in allowing us to show negative results regarding finite subsets of $\Theta$. Appendix F gives one such example.

## 4 NO RANDOMIZED ALGORITHM CAN ESTIMATE $Z$ ACCURATELY

We've shown by Theorem 2 that for an EC-complete family, there are no exact estimators than can get $Z$ perfectly right. In this section, we show that no randomized exact estimator for this is unbiased. Further, there isn't even an estimator whose bias is within some factor $\epsilon$, regardless of the variance's magnitude.

**Lemma 1.** *Let $\Theta$ be an EC-complete parametric family. There is no multiplicative factor $\epsilon \in \mathbb{Q}_{>1}$ for which every $\theta \in \Theta$ can have its partition function approximated with $\hat{\mathbf{Z}}_\epsilon(\theta)$ within a factor of $\epsilon$ — with probability greater than $2/3$. That is, we cannot have*

$$P\left( (1/\epsilon) Z_{\tilde{p}_\theta} \leq \hat{\mathbf{Z}}_\epsilon(\theta) \leq \epsilon Z_{\tilde{p}_\theta} \right) > 2/3. \tag{2}$$

*Proof sketch.* Because $\hat{\mathbf{Z}}_\epsilon$ computes an estimate in finite time, it can only use finitely many distinct random tape segments. We therefore derandomize $\hat{\mathbf{Z}}_\epsilon$ by enumerating finitely many 'random' tapes, and can always return correct answer, given the $2/3$ success probability assumption, and would be able to decide HALT, making use of distributions in $\mathcal{G}_{\epsilon^2}$. $\square$

Taken together, Theorem 2 and Lemma 1 state that no exact estimator $\hat{Z}$ — whether randomized or deterministic — can approximate $Z$ with good confidence. In Theorem 4 below, we will make an even stronger claim: regardless of the dispersion magnitude, it is impossible to bound the mean of random exact estimators of $Z$ to within any (computable) multiplicative factor. This is because the mean of $\hat{\mathbf{Z}}_\epsilon$ can be computed in finite time, by derandomizing $\hat{\mathbf{Z}}_\epsilon$ similarly to our proof of Lemma 1:

**Theorem 4.** *Let $\Theta$ be an EC-complete parametric family. There is no multiplicative bound $\epsilon \in \mathbb{Q}_{>1}$ such that there exists a randomized exact estimator $\hat{\mathbf{Z}}_\epsilon$ that guarantees $1/\epsilon \leq \mathbb{E}[\hat{\mathbf{Z}}_\epsilon(\theta)]/Z_{\tilde{p}_\theta} \leq \epsilon$, for every $\theta \in \Theta$ where $\tilde{p}_\theta$ is normalizable.*

*Proof sketch.* We derandomize the expectation of $\hat{\mathbf{Z}}_\epsilon$. $\square$

## 5 COMMON ASYMPTOTIC ESTIMATORS DO NOT GIVE USEFUL GUARANTEES

Let's now recap the progress we've made so far. We've shown that no (deterministic) exact estimator can get $Z$ exactly right in general (Theorem 2), lest it need to solve HALT. Further, no randomized exact estimator can approximate it within any given relative tolerance, with good confidence (Theorem 4).

But what about *asymptotic* estimators? We do know there are correct asymptotic estimators of $Z$. For example, consider the following asymptotic estimator $\hat{Z}(\theta)$ backed by a weighted Turing machine that takes $\theta \in \Theta$ and $i \in \mathbb{N}$ as inputs, and returns $f_{\theta,i} \triangleq \sum_{x:x \in \mathbb{B}^*, |x| \leq i} \tilde{p}_\theta(x)$. We have $\lim_{i \to \infty} \hat{Z}_{\theta,i} = Z_{\tilde{p}_\theta}$, so $\hat{Z}$ is asymptotically correct. However, $\hat{Z}_{asym}$ does not have a convergence rate guarantee: for any $i \in \mathbb{N}$, $\|\hat{Z}_{\theta,i} - Z_{\tilde{p}_\theta}\|$ is uncomputable. We also do not know how much can we improve our estimator when we increment $i$. As Corollary 2 suggests, we likely cannot have such a guarantee.

In this section, we formalize this intuition for two popular asymptotic estimators: rejection and importance sampling methods (with other asymptotic estimators left as future work). Specifically, we show that any parametric family that is able to parametrize $\tilde{b}$ from §3.2 cannot have provably useful locally normalized distributions (§2.3) as proposal distributions.

### 5.1 REJECTION SAMPLING ESTIMATOR OF $Z$ CANNOT BE GUARANTEED TO BE POSSIBLE.

Rejection sampling (Owen, 2013) is a common exact sampling method, applicable even when we cannot sample from an unnormalized distribution $\tilde{p}$. We instead sample from an easy-to-sample distribution $q$, then stochastically reject samples, to ensure the probability that a sample $x$ is kept is proportional to $\tilde{p}(x)$.

For rejection sampling to work, the candidate $q$'s support must contain the target $\tilde{p}$'s entire support, so that all true points can be sampled. We also need some finite constant $c$ so that $cq$ envelops $\tilde{p}$:

$$\exists c \in \mathbb{R}_{>0} \text{ such that } \forall x \in \mathbb{B}^*, (\tilde{p}(x)/q(x)) \leq c.$$

We will show that for certain EBMs, one cannot formally guarantee the existence of an eligible $q \in \text{LN}$.

**Theorem 5.** *Using ZFC as our axiom set and assuming they are consistent, then there exists a normalizable EC weighted language $\tilde{p}$, where there does not exist a consistent locally normalized proposal distribution $q \in \text{LN}$, and $c_q \in \mathbb{Q}_{>0}$, such that it can be proven $\forall x \in \mathbb{B}^*$, $\tilde{p}(x)/q(x) < c_q$.*

*Proof sketch.* Let $\tilde{p} = \tilde{b}$. If there were such $q$ and $c_q$, we could in finite time prove $Z_{\tilde{p}} = 1$ or $Z_{\tilde{p}} = 2$, which contradicts Theorem 3. □

Theorem 5 implies that there is no way of ensuring rejection sampling works, not only for any EC-complete families, but also for any parametric family that can parametrize $\tilde{b}$.

### 5.2 IMPORTANCE SAMPLING ESTIMATOR OF $Z$ CANNOT BE GUARANTEED TO BE EFFECTIVE.

Similar to the case of rejection sampling, one cannot guarantee an importance-sampling estimator of $Z$ to be 'good' — in this case, we mean that there cannot be a proof that the importance sampling variance is finite.

We first formalize importance sampling estimators of $Z$ as randomized asymptotic estimators (§2.5). Let

$$\hat{Z}_{\theta,N}^q = \frac{1}{N} \sum_{n=1}^{N} \frac{\tilde{p}_\theta(x^{(n)})}{q(x^{(n)})}$$

be an $N$-sample importance sampling estimator of $Z_{\tilde{p}_\theta}$ under $q$, so all $x^{(n)}$ are samples from $q \in \text{LN}$.

We generally want to minimize the variance of $\hat{Z}_{\theta,N}^q$ under $q$: $\text{Var}_q\left(\hat{Z}_{\theta,N}^q\right)$ (Owen & Zhou, 2000). And we certainly do not want $\text{Var}_q\left(\hat{Z}_{\theta,N}^q\right) = \infty$. Unfortunately, for certain EBMs, we cannot guarantee there is a good locally normalized proposal distribution that has finite variance:

**Theorem 6.** *Let $\Theta$ be an EC-complete parametric family. Assuming ZFC axioms and assuming they are consistent, there exists $\theta \in \Theta$ where there does not exist a consistent locally normalized "proposal" distribution $q \in \text{LN}$ such that it can be proven $\text{Var}_q\left(\hat{Z}_{\theta,N}^q\right) < c \neq \infty$, where $c \in \mathbb{Q}_{>0}$.*

*Proof sketch.* Proven in a manner similar to the proof of Theorem 5. □

## 6 UNCOMPUTABLE $Z$ CAUSES PARAMETER ESTIMATION PROBLEMS

Theorems 2 and 3 state that it is generally impossible to estimate partition functions in an expressive parametric family, such as certain subsets of an EC-complete family. Here we show how parameter estimation is made difficult as well: parameter identifiability is formally undecidable for an EC-complete family. Model selection is not possible, either, despite attempts to circumvent this (Table 1).

### 6.1 PARAMETER IDENTIFIABILITY UNDER EC-COMPLETE FAMILIES IS UNDECIDABLE

Consistency of many estimators relies upon the condition of parameter identifiability (Lehmann & Casella, 2006) — two different parameter vectors should define different string distributions. But we in general cannot ascertain whether this condition holds, even for finite subsets of an EC-complete family:

**Theorem 7.** *Let $\boldsymbol{\Theta}$ be an EC-complete family. There is no algorithm that takes $\boldsymbol{\theta}_1 \in \boldsymbol{\Theta}$, $\boldsymbol{\theta}_2 \in \boldsymbol{\Theta}$ as input, and decides whether $\tilde{p}_{\boldsymbol{\theta}_1}$ and $\tilde{p}_{\boldsymbol{\theta}_2}$ are the same weighted language.*

### 6.2 MODEL SELECTION IS GENERALLY IMPOSSIBLE FOR EC-COMPLETE SEQUENCE MODEL FAMILIES

Theorem 7 suggests that consistency guarantees of common parameter estimators do not hold, when used to estimate vectors from $\boldsymbol{\Theta}$. Nonetheless, such 'off-label' use of consistent parameter estimators in an expressive parametric family is quite common — one usually just selects the best model $\boldsymbol{\theta}^*$ among finitely many candidates (say $\{\boldsymbol{\theta}_1 \ldots \boldsymbol{\theta}_N\}$) that achieves highest held-out likelihood: $\boldsymbol{\theta}^* = \arg\max_{\boldsymbol{\theta}_n : 1 \le n \le N} \prod_{\boldsymbol{x} \in \mathcal{D}} p_{\boldsymbol{\theta}_n}(\boldsymbol{x})$, where $\mathcal{D}$ is a finite set of strings.

However, Theorem 8 implies that exact log-likelihood-based model selection is generally impossible for EC-complete sequence model families:

**Theorem 8.** *For any $\tilde{p} \in$ EC and for any EC-complete family $\boldsymbol{\Theta}$, there is no algorithm BETTER$_{\tilde{p}}$ that takes two parameter vectors $\boldsymbol{\theta}_1 \in \boldsymbol{\Theta}$, $\boldsymbol{\theta}_2 \in \boldsymbol{\Theta}$, and returns YES if $\mathrm{KL}(p||p_{\boldsymbol{\theta}_1}) \ge \mathrm{KL}(p||p_{\boldsymbol{\theta}_2})$, and NO otherwise, where $p$, $p_{\boldsymbol{\theta}_1}$, $p_{\boldsymbol{\theta}_2}$ are string distributions defined by $\tilde{p}$, $\tilde{p}_{\boldsymbol{\theta}_1}$, $\tilde{p}_{\boldsymbol{\theta}_2}$ respectively.*

## 7 PALLIATIVE ALTERNATIVES

What is a practitioner to do, given that this class of models is unlearnable in general? We emphasize that a model family does *not* have to be EC-complete to suffer from model selection problems (§6.2) — for example, model selection is impossible for fixed-size Transformer networks with large enough $d$'s either, by an extension to Theorem 8 (see Theorem 11 in Appendix G).[9] [10] In other words, to ensure the problem of uncomputability does not haunt us, the best we can do is to cripple $\tilde{p}$ so severely that uncomputability is impossible.

We identify three palliative choices that restrict the family of EBMs. Each cripples the model $\tilde{p}$ in its own way, affording computability at the cost of expressiveness. The choice of triage infuses the model with an inductive bias; we must tailor our models based on our prior beliefs about the problem domain.

**Restricting** support($\tilde{p}$) **to be finite.** If $\tilde{p}$ assigns non-zero weights to only finitely many strings, then $Z_{\tilde{p}}$ is a finite sum of rational numbers, and is also rational. Here, sampling-based inference and estimation methods return to their usefulness. One way to ensure support($\tilde{p}$) is finite is by upper-bounding the maximum string length (*e.g.*, Bakhtin et al. (2021)).

The finite-support restriction imposes an obvious limitation that it cannot handle long strings. Moreover, while $Z_{\tilde{p}}$ is computable when support($\tilde{p}$) is finite, this quantity can still be practically inapproximable, assuming that $\tilde{p}$ is expressive (*e.g.*, $\tilde{p}$ is an EC weighted language that has finite support), except for very short strings. Let $n$ be the longest string under $\tilde{p}$ to have non-zero weight. Assuming NP $\not\subseteq$ P/poly, no randomized estimator of $Z_{\tilde{p}}$ that is a good approximation can have a guaranteed $O(\mathrm{poly}(n))$ time complexity (Chandrasekaran et al., 2008). However, if $\tilde{p}$ has limited expressiveness (*e.g.*, when $\tilde{p}$ is an Ising model where a string weight is the sum of pairwise weights), then FPRAS algorithms for approximating $Z_{\tilde{p}}$

---

[9]In addition to model selection issues, it may also be difficult to acquire unbiased gradients of $\log Z$: $\nabla (\log Z) \triangleq \frac{1}{Z} \nabla Z$, which are needed for MLE-based training.

[10]Limiting ourselves to small $d$'s to avoid uncomputability issues may not be practical; we leave finding the largest $d$ that provably does not involve uncomputability problems — if it is even possible — as future work.

may exist when $\tilde{p}$ describe a low-degree ($\leq 2$) graph (Jerrum & Sinclair, 1993; Luby & Vigoda, 1999). However, for high-degree graphs ($\geq 3$) it can be shown no FPRAS algorithm for approximating $Z_{\tilde{p}}$ exists, assuming RP $\neq$ NP (Galanis et al., 2016).

**Autoregressive parametrization of $\tilde{p}$.** An alternative choice is to confine ourselves to autoregressive models, *i.e.*, locally normalized string distributions (§2.3). Under an autoregressive model $p$, $Z_p = 1$ by definition. We also note that any (unnormalized) distribution $\tilde{p}$ obtained by removing probability mass from $p$ will have a *computable* partition function, as long as $\tilde{p} \in$ EC:

**Theorem 9.** *Let $p$ be any* LN *weighted language. Any $\tilde{p} \in$ EC where $\forall x \in V^*$, $\tilde{p}(x) \leq p(x)$ has a computable $Z_{\tilde{p}}$.*

*Proof sketch.* We construct an algorithm that approximates $Z_{\tilde{p}}$ to any arbitrary error level in finite time, exploiting the consistency property of $p$ (Proposition 1). □

Theorem 9 implies that conditionalization operations on $p$, which remove strings from the support of $p$ to get weighted language $\tilde{p}$, result in a computable $Z_{\tilde{p}}$ (as long as we can decide which strings are removed); and such a $\tilde{p}$ is therefore not subjected to the limitations of Theorem 4.

Unlike the 'finite support' fix, an autoregressively parametrized (or subsequently conditionalized) $\tilde{p}$ can have an infinite support. A conditionalized $\tilde{p}$ can have an intractable (but computable) partition function, and they are still subject to the expressive limitations imposed on LN languages: namely there is an EC language whose string weight rankings cannot be honored by any such conditionalized $\tilde{p}$ (Lin et al., 2021).

**$\tilde{p}$ as low-treewidth factor graph grammars.** Finally, we may limit ourselves to weighted languages defined by low-treewidth factor graph grammars (Chiang & Riley, 2020). Factor graph grammars generalize factor graphs, which cover many classes of graphical sequence models, such as $n$-gram, HMM, and whole-sentence language models (Jelinek, 1980; Kuhn et al., 1994; Rosenfeld et al., 2001), linear CRF models (Lafferty et al., 2001), and weighted FSAs in general (Dreyer & Eisner, 2009): a factor graph grammar describes a (possibly infinite) set of factor graphs, generated from a hyperedge replacement graph grammar.

Assuming that an FGG $G$ contains at most one $n$-observed-node factor graph for all $n \in \mathbb{N}$, it then defines a weighted language $\tilde{p}_G(x) = \prod_{\psi \in \Psi_{|x|}} \psi$, where **factor** $\psi$ is a positive function of nodes. The treewidth of an FGG $W(G)$ is the maximum number of nodes any $\psi$ can be a function of, and $\Psi_{|x|}$ is the set of all factors of string length $|x|$.

If $Z_{\tilde{p}_G} \in \mathbb{R}$ exists, it can be computed exactly by an algorithm, in time exponential in $W(G)$ following Chiang & Riley (2020). Exact computation of $Z_{\tilde{p}_G}$ may be manageable as long as $W(G)$ is small, which would allow us to exactly compute held-out data likelihood, and also train with a (marginalized) MLE objective function. However, limiting $W(G)$ directly limits the expressiveness of $\tilde{p}$.

## 8 RELATED WORK

Turing completeness of formal languages and associated uncomputability issues emerge repeatedly in computer science. For example, Turing completeness may emerge as an unwanted side effect in programming languages, since it implies undecidability. One of the best known examples is the Turing completeness of the C++ grammar (Veldhuizen, 2003; Haberman, 2013), which makes both parsing and compiling C++ programs undecidable. Similar problems exist for Java (Grigore, 2016) and Haskell with (unrestricted) instance declarations (Wansbrough, 1998). Another example is the (possibly inadvertently introduced) Turing completeness of the page fault handling mechanism on modern computer systems, which raises security concerns in the context of trusted computing (Bangert et al., 2013).

Our work is not the first to discuss the consequences of computability in machine learning: assuming we can acquire training data from an oracle, under a supervised learning setting, recognition of an undecidable language is PAC-learnable (Lathrop, 1996). Agarwal et al. (2020) extended the definition of PAC learnability (Valiant, 1984) to computable learners. By contrast, we are focused on the computability of EBMs for sequences, such as a language model as a component of a larger system for automatic speech recognition. Designing an appropriate, efficient loss functional is a challenge that several prior works have compared. With the plethora of learning strategies for EBMs, it is untenable to point out the

| Technique | Energy-based (*i.e.*, globally normalized) | Infinite language support | Scoring function has unbounded treewidth | Consistency |
|---|---|---|---|---|
| Noise-contrastive estimation (Ma & Collins (2018); used in Lin et al. (2021); Bakhtin et al. (2021)) | ✓ | ✗ | ✓ | ✓ |
| MLE with variable elimination in factor graph grammars (Chiang & Riley (2020); used in Eisner (2001); Finkel et al. (2008), *inter alia*) | ✓ | ✓ | ✗ | ✓ |
| MLE with autoregressive parametrization (Mikolov et al., 2010) | ✗ | ✓ | ✓ | ✓ |
| Contrastive divergence (Hinton, 2002) | ✓ | ✓ | ✓ | ✗ |
| Contrastive estimation (Smith & Eisner, 2005) | ✓ | ✓ | ✓ | ✗ |

Table 1: Deficiencies of some common alternatives to overly expressive EBMs.

deficiency in each. Table 1 gives a handful of examples; none share the four properties we desire: 1. Global normalization (without which the model would be in LN) 2. Support over infinite languages (of finite strings) 3. Unbounded treewidth in the function assigning weights to strings 4. Estimator consistency (*i.e.*, asymptotic guarantee to recover the true parameters).

Lin et al. (2021) noted autoregressive factors of EC languages can be uncomputable (see also Theorem 10). They also noted that a weighted language can have an uncomputable partition function (presumably resulting from the sum over infinitely many string weights). But they did not dwell on the question whether such a weighted language could lie within the EC class, much less providing a constructive proof (see also Appendix E). Instead, they emphasized that under the assumption that oracular access to trained parameter strings is possible, arbitrarily good approximations of the (possibly uncomputable) partition function can be memorized in the (autoregressive) model parameters. There is an interesting constrast between the stances of Lin et al. (2021) and our work: Lin et al. (2021) saw the uncomputability of $Z$ as a trivial issue from the model capacity viewpoint, since good approximations take few bits in the parameters to store. On the other hand, we see that the uncomputability problem can be a parameter estimation disaster — there will be no guarantee of good approximations can be found in finite time at all.

## 9    CONCLUSION AND FUTURE WORK

Energy-based models are posed as an efficient tool for decision problems, circumventing probabilities and expensive normalization (LeCun et al., 2006). Extending this vision to generic sequence models, though, can involve complexity/computability problems that are difficult, or even impossible. We've shown that as energy-based sequence models become more powerful, the partition function becomes uncomputable — even when we are restricted to polytime-computable weighted languages. Exact estimators, even if randomized, cannot have accuracy guarantees. Popular asymptotic estimators on the other hand are not useful either. Furthermore, model selection is generally impossible, even if we limit ourselves to fixed-size sequence model families.

This paper continues a discussion started by Lin et al. (2021), who posture energy-based models as a more powerful alternative to autoregressive sequence models. Autoregressive sequence models, after all, have wide adoption and empirical successes (Radford et al., 2019; Brown et al., 2020). By contrast, more general neural energy-based sequence models have not caught on. Why not? We give *unlearnability* — due to *uncomputability* — as a possible explanation: unless we give up the ability to learn parameters from data, we likely cannot use the full expressiveness afforded by powerful neural networks. Just like the model capacity problems brought up by Lin et al. (2021), this result is independent of the amount of training data.

We emphasize that our results do not invalidate the findings of Lin et al. (2021): regardless of the actual neural parametrization, autoregressive models can never capture certain distributions that energy-based models can. Instead, one of our main messages is that *we may not be able to find those EBM parameters in finite time, if we do not know what the parameters are*. Of course, if we know the task perfectly well and can in fact manually assign the model parameters, we will not need to learn from data at all. The middle ground — when we have *some* prior knowledge about the task, but cannot really design the parameter vectors — is an interesting direction for future work: the three palliative alternatives outlined in §7 do not take task-specific information into account at all. Can we do better than that, without suffering uncomputability problems?

## ACKNOWLEDGMENTS

We thank the four reviewers for their comments, especially Reviewer NAcm for shortening the proof of Theorem 4. Additionally, we thank Alexandra DeLucia, Matthew Francis-Landau, Chin-Fu Liu, Suzanna Sia, Neha Verma, and Chenghao Yang (sorted alphabetically) for discussions that improved the presentation; and Jason Eisner, discussions with whom motivated the original exploration of this work.

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

|  | exact | asymptotic |
|---|---|---|
| deterministic | ✗ (Theorem 2) | ✓ (§5; but no useful guarantee in finite time) |
| randomized | ✗ (Theorem 4) | ?; but ✗ for rejection sampling (Theorem 5) and ✗ for importance sampling (Theorem 6) when paired with autoregressive proposal distributions |

Table A.1: Summary of negative results: neither deterministic or randomized algorithms can estimate EBM partition functions accurately. On the other hand, popular sampling schemes such as rejection and importance sampling require their autoregressive proposal distributions to be uncomputable.

# A   Summary of negative results

In Table A.1, we give an overview of the negative results in this paper: deficiencies of several estimators of the partition function in an energy-based sequence model.

# B   Weighted Turing machines

We extend the definition of Turing machines into **weighted Turing machines** as follows. A weighted Turing machine $M$ is described as a 7-tuple $(Q, \Sigma, \delta, q_{\text{init}}, F, \texttt{update\_num}, \texttt{update\_denom})$, where the first 5 components $(Q, \Sigma, \delta, q_{\text{init}}, F)$ have definitions following that of standard Turing machines. The last two additions are weight-update functions: both `update_num` and `update_denom` have signature $Q \rightarrow \mathcal{A}$, where $\mathcal{A} \triangleq \{\text{SAME}, \text{CARRY}\}$.

At the end of time step $i$, assuming the current state is $q$, we define

$$\text{NUM}_i = \begin{cases} \text{NUM}_{i-1} & \text{if } \texttt{update\_num}(q) = \text{SAME} \\ 2^i + \text{NUM}_{i-1} & \text{if } \texttt{update\_num}(q) = \text{CARRY} \end{cases}$$

and similarly,

$$\text{DENOM}_i = \begin{cases} \text{DENOM}_{i-1} & \text{if } \texttt{update\_denom}(q) = \text{SAME} \\ 2^i + \text{DENOM}_{i-1} & \text{if } \texttt{update\_denom}(q) = \text{CARRY}. \end{cases}$$

To keep our proof of model family capacity brief, we (arbitrarily) define that $\forall q \in F \cup \{q_{\text{init}}\}$, $\texttt{update\_num}(q) = \texttt{update\_denom}(q) = \text{SAME}$. Finally, upon arriving at a halting state $\in F$ in $r$ time steps, we say $\text{NUM}_r / \text{DENOM}_r$ is the weight of an input $x \in \mathbb{B}^*$ under $M$, if $\text{NUM}_r / \text{DENOM}_r$ is a rational number.[11]

# C   Definition of sequence model families and EC-complete parametric families

Following Pérez et al. (2021), we say a **sequence model family** is a set of **seq-to-seq** models $\mathbf{N} = \{N_\theta : \theta \in \Theta \subseteq \mathbb{Q}^*\}$, where every $N_\theta \in \mathbf{N}$ **recognizes** language $L_\theta$ if and only if the seq-to-seq model $N_\theta$, paired with **embedding function** $f : \mathbb{B} \rightarrow \mathbb{Q}^d$, **initial states** $s \in \mathbb{Q}^d$, and polytime **termination decision function** $g : \mathbb{Q}^d \rightarrow \mathbb{B}$, *accepts* every string $x \in L_\theta$. And we define $N_\theta$ to **accept** $x = [x_1 \dots x_T] \in \mathbb{B}^*$ if and only if there exists $r \in \mathbb{N}$, such that $N_\theta$ upon input embeddings $[f(x_1) \dots f(x_T)]$, produces output embeddings $y_r$ at the $r$-th time step, where $g(y_r) = 1$.

A sequence family $\Theta$ is EC-complete if given any $\tilde{p} \in \text{EC}$ (as a description of a weighted Turing machine), we can construct a parameter vector $\theta \in \Theta$ such that there exist polytime functions $w_p : \mathbb{Q}^d \rightarrow \mathbb{N} \cup \{0\}$ and $w_q : \mathbb{Q}^d \rightarrow \mathbb{N} \cup \{0\}$, where whenever on input $x = [x_1 \dots x_T]$, if $g(y_r) = 1$ for some output embeddings $y_r$ (that is, $N_\theta$ accepts $x$ in $r$ time steps), $w_P(y_r) / w_q(y_r) = \tilde{p}(x)$.

---

[11]When $\text{NUM}_r / \text{DENOM}_r$ is not a rational number (because $\text{DENOM}_r = 0$), we say the weight of input string $x$ is undefined. However, we do not encounter that in this work: all strings in an EC language have rational weights. In any case, we only require that our family *contain* weighted Turing machines that return rational numbers, not that they be the only members.

# D    PROOFS OF THEOREMS FROM MAIN TEXT

**Theorem 1.** *The class of one-encoder-layer, four-decoder-layer Transformer networks with positional encodings $(n, 1/n, 1/n^2, 2^n)$ is* EC-*complete.*

*Proof.* To model all weighted Turing machines defined in §2.2, we extend the Turing-complete one-encoder-layer three-decoder-layer Transformer network construction introduced by Pérez et al. (2021) with one additional layer. We also modify the original positional embedding function pos : $\mathbb{N} \rightarrow \mathbb{Q}^d$ to include a new component. In this proof, we let

$$\text{pos}(i) = [0, \ldots, 0, 1, i, 1/i, 1/i^2, 2^i]$$

instead of $\text{pos}(i) = [0, \ldots, 0, 1, i, 1/i, 1/i^2]$ as in Pérez et al. (2021).

For the sake of clarity, our construction is a 'stack-on' construction: the encoder layer and the first 3 decoder layers are largely identical to the design of Pérez et al. (2021), with the only difference being necessary changes to accommodate our one additional positional embeddings component.[12] It may be possible to strengthen our results by showing that one-encoder-layer **three**-decoder-layer Transformer networks with the original positional embeddings — the parametrization family Pérez et al. (2021) showed to be Turing-complete — are EC-complete as well, with a more involved construction. We leave such an improvement as future work.

We claim our fourth layer of the decoder has output $\boldsymbol{y}_r = \boldsymbol{y}_r' + \boldsymbol{w}_r$, where

- $\boldsymbol{y}_r'$ is a zero-padded version of the original Transformer output embeddings from Pérez et al. (2021). $\boldsymbol{y}_r'$ is of the form

$$[\llbracket q^r \rrbracket, \llbracket s^r \rrbracket, m^{(r-1)}, 0, \ldots, 0]$$

  where $\llbracket q^r \rrbracket$ denotes an one-hot vector of size $|Q|$ where the $q^r$-th component is 1 (again following the notation of Pérez et al. (2021)).
- And $\boldsymbol{w}_r$ is of the form

$$[\boldsymbol{0}_q, \boldsymbol{0}_s, 0, \text{NUM}_r, \text{DENOM}_r, 0, \ldots, 0] \qquad (3)$$

  where $\text{num}_r \in \mathbb{N} \cup \{0\}$ and $\text{DENOM}_r \in \mathbb{N} \cup \{0\}$ are defined in §2.2.

Now we describe how $\boldsymbol{w}_r$ can be computed from $[\boldsymbol{y}_0' \ldots \boldsymbol{y}_r']$ using the attention mechanism, with the help of our new positional embeddings. Specifically, we want to show that we can construct feedforward networks $Q_4$, $K_4$, and $V_4$ such that

$$\boldsymbol{y}_i = \text{Att}\big(Q_4(\boldsymbol{y}_i''), K_4(\boldsymbol{Y}_i''), V_4(\boldsymbol{Y}_i'')\big)$$

where $\boldsymbol{y}_i'' = \boldsymbol{y}_i' + \text{pos}(i)$, $\boldsymbol{Y}' = [\boldsymbol{y}_1' \ldots \boldsymbol{y}_i']$, and $\boldsymbol{Y}_i'' = \boldsymbol{Y}' + [\text{pos}(1), \ldots, \text{pos}(i)]$. We let $Q_4(\boldsymbol{y}'') = [0, \ldots, 0, 1, 0, 0, 0, 0]$ be a constant vector, and $K_4(\boldsymbol{Y}_i'') = \boldsymbol{Y}_i''$. Finally, we let $V_4(\boldsymbol{Y}_i'') = [\boldsymbol{0}_q, \boldsymbol{0}_s, 0, \mathbb{I}(\texttt{update\_num}(q^i) = \text{CARRY})2^i, \mathbb{I}(\texttt{update\_denom}(q^i) = \text{CARRY})2^i, 0, \ldots, 0]$.

$Q_4$ is a constant function (such that it always attends to the unity component of $\boldsymbol{Y}_i''$), so it can be implemented as a single-layer feedforward network. $K_4$ is the identity function, which can also be implemented as a single-layer feedforward network. $V_4$ on the hand can be implemented as a fixed-size feedforward network, with the piecewise-linear sigmoidal function $\sigma$: in the case of $\text{DENOM}_i$, the network would first project $q^i$ to $\boldsymbol{a} = [\mathbb{I}(\texttt{update\_denom}(q^i) = \text{CARRY}), \mathbb{I}(\texttt{update\_denom}(q^i) = \text{SAME})]$ (using the one-hot $\llbracket q^i \rrbracket$ segment from $\boldsymbol{y}_i'$), multiply it by $\boldsymbol{b} = [2^i, 0]$ (with the help of nonlinearity from $\sigma$), and put $\boldsymbol{a} \otimes \boldsymbol{b}$ at the position of $\text{DENOM}_i$ in equation (3). The component at $\text{NUM}_i$ in equation (3) can be computed likewise.

---

[12]Since we increase the output embeddings' dimension by 1, we also need to pad all matrices in the original construction by additional zero columns/rows, such that our new positional embeddings' new component has no effect on any computation in the encoder layer, and the first 3 decoder layers.

Given any position $r \in \mathbb{N}$, we have

$$
y_r = \text{Att}(Q_4(y_r''), K_4(\mathbf{Y}_r''), V_4(\mathbf{Y}_r'')) = [\mathbf{0}_q,
$$
$$
\mathbf{0}_s,
$$
$$
0,
$$
$$
\frac{1}{r} \sum_{i=1}^{r} \mathbb{I}(\texttt{update\_num}(q^i) = \text{CARRY})2^i,
$$
$$
\frac{1}{r} \sum_{i=1}^{r} \mathbb{I}(\texttt{update\_denom}(q^i) = \text{CARRY})2^i,
$$
$$
0,
$$
$$
\dots,
$$
$$
0].
$$

Let $\texttt{extract\_avg\_num}(y_r)$ be an affine transformation that extracts the $(|Q| + |\Sigma| + 2)^{\text{nd}}$ component from $y_r$, and $\texttt{extract\_avg\_denom}(y_r)$ be an affine transformation that extracts the $(|Q| + |\Sigma| + 3)^{\text{rd}}$ component from $y_r$, we have

$$
\frac{\texttt{extract\_avg\_num}(y_r)}{\texttt{extract\_avg\_denom}(y_r)} = \frac{\sum_{i=1}^{r} \mathbb{I}(\texttt{update\_num}(q^i) = \text{CARRY})2^i}{\sum_{i=1}^{r} \mathbb{I}(\texttt{update\_denom}(q^i) = \text{CARRY})2^i}
$$
$$
= \frac{\text{NUM}_r}{\text{DENOM}_r}
$$

which is the weight of input $x = [x_1 \dots x_T]$ as we defined in §2.2. $\qquad \square$

**Theorem 2.** *Let $\Theta$ be a parametric* EC*-complete sequence model family. And let $\Theta_k \subset \Theta$ be bijective with $\mathcal{G}_k$. There is no $k \in \mathbb{Q}_{>0}$ for which there exists an exact estimator $\hat{Z}_k$ that takes as input $\theta \in \Theta_k$ as input, and computes $\hat{Z}_k(\theta) = Z_{\tilde{p}_\theta}$ in finite time.*

*Proof.* We can reduce HALT to computing our partition function. For the sake of contradiction, let us assume that for some $k \in \mathbb{Q}_{>0}$, the exact estimator $\hat{Z}_k$ exists. Our reduction from HALT of input-free Turing machines is as follows: Given any deterministic input-free Turing machine $M$, we build a *weighted deterministic Turing machine*: $M'$ (Appendix B). $M'$ takes as input $x \in \mathbb{B}^*$, and outputs weight $1/3^{|x|+1} + k$ when $x$ encodes an accepting trace of $M'$. Otherwise, $M'$ outputs weight $1/3^{|x|+1}$.

$M'$ always returns a weight for $x$ in polytime. By the assumptions of EC-complete families (§2.4), we can build a parameter vector $\theta \in \Theta$ such that $\tilde{p}_{M'} = \tilde{p}_\theta$. Since from the definitions of $\mathcal{G}_k$ we know $\tilde{p}_{M'} = \tilde{p}_{M,k}$, we have $\theta \in \Theta_k$.

We have thus completed our reduction: if $\hat{Z}_k$ existed, we could decide whether any given input-less deterministic Turing machine $M$ halts, by first constructing the weighted Turing machine $M'$, then the corresponding $\theta$. By our assumption, $\hat{Z}_k(\tilde{p}_\theta) = k + 1$ if and only if $\exists x \in \mathbb{B}^*$ that is an accepting path of $M'$, which is true if and only if $M$ halts for some finite steps. Since whether $M$ halts is undecidable, we have arrived at a contradiction (Turing, 1937; Sipser, 2013). Therefore for all $k \in \mathbb{Q}$, the algorithm $\hat{Z}_k$ does not exist. $\qquad \square$

**Theorem 3.** *Assuming ZFC axioms and that assuming they is consistent, there exists an* EC *weighted language $\tilde{b} \in \mathcal{G}_1$ such that (a) $Z_{\tilde{b}} = c \in \{1, 2\}$ exists, but (b) there is no proof of $Z_{\tilde{b}} = c$.*

*Proof.* Our proof hinges on Gödel's second incompleteness theorem: no consistent axiomatic system which includes Peano arithmetic (*e.g.*, ZFC) can prove its own consistency. We construct a Turing machine $M_b$ that enumerates all provable propositions under ZF, and halts if and only if $M_b$ proves the proposition $1 = 0$. One such $M_b$ with $7,918$ states has been built by Yedidia & Aaronson (2016).[13]

$M_b$ is an input-free deterministic Turing machine. We construct a weighted Turing machine $M'$ from $M_b$,

---

[13]Subsequent efforts have led to a construction of $M_b$ with 748 states: https://turingmachinesimulator.com/shared/vgimygpuwi.

in the manner of the proof of Theorem 2. We let $M'$ return weight $1 + 1/3^{|\boldsymbol{x}|+1}$, in the case $M_b$ halts with trace $\boldsymbol{x}$. $M'$ returns a weight in polytime, and therefore defines a weighted language $\tilde{b} \in \mathcal{G}_1 \subset$ EC. We know from the definitions of $\mathcal{G}_1$ that $Z_{\tilde{b}}$ is either 1 or 2.

Assume to the contrary that there exists a proof that $Z_{\tilde{b}} = 1$. Then we know $M_b$ did not halt. And therefore the proof would also imply that ZFC is consistent, which violates Gödel's second incompleteness theorem. On the other hand, if there were a proof that $Z_{\tilde{b}} = 2$, it would imply $M_b$ halted and ZFC is not consistent. We therefore arrive at a contradiction. $\qquad\square$

**Lemma 1.** *Let* $\boldsymbol{\Theta}$ *be an* EC-*complete parametric family. There is no multiplicative factor* $\epsilon \in \mathbb{Q}_{>1}$ *for which every* $\boldsymbol{\theta} \in \boldsymbol{\Theta}$ *can have its partition function approximated with* $\hat{\mathbf{Z}}_\epsilon(\boldsymbol{\theta})$ *within a factor of* $\epsilon$ *— with probability greater than* 2/3. *That is, we cannot have*

$$P\left( (1/\epsilon)\, Z_{\tilde{p}_{\boldsymbol{\theta}}} \le \hat{\mathbf{Z}}_\epsilon(\boldsymbol{\theta}) \le \epsilon Z_{\tilde{p}_{\boldsymbol{\theta}}} \right) > 2/3. \tag{2}$$

*Proof.* In this proof, we make use of the distribution family $\mathcal{G}_{\epsilon^2}$ (§3.1). We assume to the contrary that a multiplicative bound $\epsilon$ satisfying equation (2) exists. Recall that our assumptions state that $P\left( 1/\epsilon \le \hat{\mathbf{Z}}_\epsilon(\boldsymbol{\theta})/Z_{\tilde{p}_{\boldsymbol{\theta}}} \le \epsilon \right) > 2/3.$[14] Let $M$ be an input-free Turing machine. And let $\tilde{p}_{\boldsymbol{\theta}} = \tilde{p}_{M,\epsilon^2} \in \mathcal{G}_{\epsilon^2}$ where $\boldsymbol{\theta} \in \boldsymbol{\Theta}$. If the Turing machine $M$ halts, $Z_{\tilde{p}_{M,\epsilon^2}} = 1 + \epsilon^2$; therefore, $P\left( (1 + \epsilon^2) 1/\epsilon \le \hat{\mathbf{Z}}_\epsilon(\boldsymbol{\theta}) \le (1 + \epsilon^2)\epsilon \right) >$ 2/3. Similarly, if $M$ does not halt, we have $Z_{\tilde{p}_{M,\epsilon^2}} = 1$; therefore, $P\left( 1/\epsilon \le \hat{\mathbf{Z}}_\epsilon(\boldsymbol{\theta}) \le \epsilon \right) > $ 2/3. By combining the two conditions, we know that $P\left(\mathbb{I}(M \text{ halts})\right) = P\left( \mathbb{I}(\hat{\mathbf{Z}}_\epsilon(\boldsymbol{\theta}) \ge \epsilon + 1/\epsilon \wedge \hat{\mathbf{Z}}_\epsilon(\boldsymbol{\theta}) > \epsilon) \right) = P\left( \mathbb{I}(\hat{\mathbf{Z}}_\epsilon(\boldsymbol{\theta}) \ge \epsilon + 1/\epsilon) \right)$.

Therefore, given $(1/\epsilon)\, Z_{\tilde{p}_{\boldsymbol{\theta}}} \le \hat{\mathbf{Z}}_\epsilon(\boldsymbol{\theta}) \le \epsilon Z_{\tilde{p}_{\boldsymbol{\theta}}}$, we can decide if $M$ halts by checking if $\hat{\mathbf{Z}}_\epsilon(\boldsymbol{\theta}) \ge \epsilon + 1/\epsilon$. Since the condition $(1/\epsilon)\, Z_{\tilde{p}_{\boldsymbol{\theta}}} \le \hat{\mathbf{Z}}_\epsilon(\boldsymbol{\theta}) \le \epsilon Z_{\tilde{p}_{\boldsymbol{\theta}}}$ only holds 2/3 of all time, we then derandomize the randomized $\hat{\mathbf{Z}}_\epsilon$ to get a deterministic estimator: recall that a randomized exact estimator finishes computation in finite time, regardless of the content of the random tape $\tau$. There, there are only finitely many finitely long possible 'random' sequences that $\hat{Z}_\epsilon$ will read, which we can enumerate. More concretely:

$$\begin{aligned} P\left(\hat{\mathbf{Z}}_\epsilon(\boldsymbol{\theta}) \ge \epsilon + 1/\epsilon\right) &= \mathbb{E}_\tau\left[ \mathbb{I}(\hat{Z}_\epsilon(\boldsymbol{\theta}, \tau) \ge \epsilon + 1/\epsilon) \right] \\ &= \sum_{\tau \in \mathbb{B}^{m_{\boldsymbol{\theta},\epsilon}}} 1/2^{m_{\boldsymbol{\theta},\epsilon}} \mathbb{I}(\hat{Z}_\epsilon(\boldsymbol{\theta}, \tau) \ge \epsilon + 1/\epsilon) \end{aligned} \tag{4}$$

where $m_{\boldsymbol{\theta},\epsilon} \in \mathbb{N}$ is the maximum prefix length of the random tape that $\hat{Z}_\epsilon(\boldsymbol{\theta}, \tau)$ will use on any $\tau \in \mathbb{B}^{\mathbb{N}}$. Again $m_{\boldsymbol{\theta},\epsilon}$ is guaranteed to exist because of our assumption $\hat{Z}_\epsilon(\boldsymbol{\theta}, \tau)$ ends in finite time. Since equation (4) is a finite sum of computable functions, it is also computable.

From the computability of equation (4), it follows that we can derive a deterministic algorithm from $\hat{\mathbf{Z}}_\epsilon$ that decides whether an arbitrary input-free Turing machine halts, which is not possible. Therefore, there is no $\epsilon \in \mathbb{Q}_{>1}$, such that the randomized exact estimator $\hat{\mathbf{Z}}_\epsilon$ satisfies the multiplicative bound $P(1/\epsilon \le \hat{\mathbf{Z}}_\epsilon(G)/Z \le \epsilon) > 2/3$ for all $\boldsymbol{\theta} \in \boldsymbol{\Theta}$. $\qquad\square$

**Theorem 4.** *Let* $\boldsymbol{\Theta}$ *be an* EC-*complete parametric family. There is no multiplicative bound* $\epsilon \in \mathbb{Q}_{>1}$ *such that there exists a randomized exact estimator* $\hat{\mathbf{Z}}_\epsilon$ *that guarantees* $1/\epsilon \le \mathbb{E}[\hat{\mathbf{Z}}_\epsilon(\boldsymbol{\theta})]/Z_{\tilde{p}_{\boldsymbol{\theta}}} \le \epsilon$, *for every* $\boldsymbol{\theta} \in \boldsymbol{\Theta}$ *where* $\tilde{p}_{\boldsymbol{\theta}}$ *is normalizable.*

*Proof.* We define $\tau$ to be distributed according $p_\tau$. Therefore the mean $\mathbb{E}[\hat{\mathbf{Z}}_\epsilon(\boldsymbol{\theta})]$ can be expanded as $\mathbb{E}_{\tau \sim p_\tau}[\hat{\mathbf{Z}}_\epsilon(\boldsymbol{\theta})] = \sum_{\tau \in \mathbb{B}^{\mathbb{N}}} p_\tau(\tau) \hat{Z}_\epsilon(\boldsymbol{\theta}, \tau)$. Following the same derandomization technique in the proof of Lemma 1, we can find some $m \in \mathbb{N}$ such that $\sum_{\tau \in \mathbb{B}^{\mathbb{N}}} p_\tau(\tau) \hat{Z}_\epsilon(\boldsymbol{\theta}, \tau) = \sum_{\tau \in \mathbb{B}^m} 1/2^m \hat{Z}_\epsilon(\boldsymbol{\theta}, \tau)$ in finite time. And subsequently, we can compute $\mathbb{E}[\hat{\mathbf{Z}}_\epsilon(\boldsymbol{\theta})]$ exactly in finite time. Let the exact estimator be $\overline{Z}_\epsilon$. We then have $\overline{Z}_\epsilon(\boldsymbol{\theta}) = \mathbb{E}[\hat{\mathbf{Z}}_\epsilon(\boldsymbol{\theta})]$.

---

[14]As is common, *e.g.*, in defining the complexity class BPP (see Arora & Barak 2006), the fraction 2/3 is arbitrary. Any proportion bounded away from 1/2 will work.

Assuming to the contrary that we could guarantee $1/\epsilon \leq \overline{Z}_\epsilon(\boldsymbol{\theta}) \leq \epsilon$. Since $\overline{Z}_\epsilon$ is a deterministic estimator, we can write $P(1/\epsilon \leq \overline{Z}_\epsilon(\boldsymbol{\theta}) \leq \epsilon) = 1$, which contradicts Lemma 1. Therefore such an estimator $\overline{Z}_\epsilon(\boldsymbol{\theta})$ does not exist. $\qquad\square$

**Theorem 5.** *Using ZFC as our axiom set and assuming they are consistent, then there exists a normalizable* EC *weighted language $\tilde{p}$, where there does not exist a consistent locally normalized proposal distribution $q \in$ LN, and $c_q \in \mathbb{Q}_{>0}$, such that it can be proven $\forall \boldsymbol{x} \in \mathbb{B}^*$, $\tilde{p}(\boldsymbol{x})/q(\boldsymbol{x}) < c_q$.*

*Proof.* By contradiction; let $\tilde{p} = \tilde{b}$, where $\tilde{b}$ is first introduced in §3.2. Assume that there exists $q \in$ LN where it is proven that $\forall \boldsymbol{x} \in \mathbb{B}^*$, $\tilde{p}(\boldsymbol{x})/q(\boldsymbol{x}) < c < \infty$, where $c \in \mathbb{Q}_{>0}$.

We can either prove ZFC is consistent, or is inconsistent, as follows:

1. By our assumptions, $q \in$ LN scores every string $\boldsymbol{x}$ as $q_{\boldsymbol{\theta}}(\boldsymbol{x})f(\boldsymbol{\theta}) \in \mathbb{Q}_{>0}$. Also, because $q$ is consistent and locally normalized, there exists $n \in \mathbb{N}$ such that we can prove $\forall \boldsymbol{x} \in \mathcal{X}_{>n}\{\boldsymbol{x} : \boldsymbol{x} \in \mathbb{B}^*, |\boldsymbol{x}| > n\}, q(\boldsymbol{x}) < 1/c$ (Proposition 1). We will just prove the existence of $n$ constructively by enumerating the strings in $\mathbb{B}^*$ in increasing length. Let the complement set $\mathcal{X}_{\leq n} = \mathbb{B}^* - \mathcal{X}_{>n}$.
2. The proof then examines the finitely many strings in $\mathcal{X}_{\leq n}$.
   (a) If any of these strings $\boldsymbol{x}'$ has $\tilde{b}(\boldsymbol{x}') > 1$, then we know $\boldsymbol{x}'$ encodes an accepting trace of $M_b$ (§3.2). Therefore, $M_b$ halts, which implies there is a proof of the inconsistency of ZFC.
   (b) If none of these strings $\in \mathcal{X}_{\leq n}$ has $\tilde{b}(\boldsymbol{x}) > 1$, then we know there is also no string $\boldsymbol{x}'' \in \mathcal{X}_{>n}$, such that $\tilde{b}(\boldsymbol{x}'') > 1$. This is because of our assumption $c \geq \tilde{b}(\boldsymbol{x})/q(\boldsymbol{x})$, which in turn means that $\tilde{b}(\boldsymbol{x})$ is less than 1 on these long strings $\mathcal{X}_{>n}$. Therefore, $M_b$ does not halt, which implies there is a proof of the consistency of ZFC.

Assuming ZFC is consistent, neither a proof of ZFC, or a disproof of ZFC, is possible. We have therefore therefore arrived at a contradiction. Therefore, $q$ does not exist. $\qquad\square$

**Theorem 6.** *Let $\boldsymbol{\Theta}$ be an* EC*-complete parametric family. Assuming ZFC axioms and assuming they are consistent, there exists $\boldsymbol{\theta} \in \boldsymbol{\Theta}$ where there does not exist a consistent locally normalized "proposal" distribution $q \in$ LN such that it can be proven $\mathrm{Var}_q\left(\hat{\mathbf{Z}}^q_{\boldsymbol{\theta},N}\right) < c \neq \infty$, where $c \in \mathbb{Q}_{>0}$.*

*Proof.* let $\tilde{p} = \tilde{b}$, where $\tilde{b}$ is first introduced in §3.2. Assume that there exists $q \in$ LN where it is proven that $\mathrm{Var}_q(\hat{\mathbf{Z}}^q_{\boldsymbol{\theta},N}) = k < c \in \mathbb{Q}_{<0} \neq \infty$.

We have

$$
\begin{aligned}
&\mathrm{Var}_q(\hat{\mathbf{Z}}^q_{\boldsymbol{\theta},N}) \\
&\triangleq \frac{1}{N}\left\{\sum_{\boldsymbol{x} \in \mathbb{B}^*}\left[\left(\frac{\tilde{p}_{\boldsymbol{\theta}}(\boldsymbol{x})}{q(\boldsymbol{x})}\right)^2 q(\boldsymbol{x})\right] - Z_{\tilde{p}_{\boldsymbol{\theta}}}{}^2\right\} \\
&= \frac{Z^2_{\tilde{p}_{\boldsymbol{\theta}}}}{N}\left(\sum_{\boldsymbol{x} \in \mathbb{B}^*}\frac{p^2_{\boldsymbol{\theta}}(\boldsymbol{x})}{q(\boldsymbol{x})} - 1\right) \\
&= k,
\end{aligned}
$$

where $p_{\boldsymbol{\theta}}(\boldsymbol{x}) \triangleq \frac{\tilde{p}_{\boldsymbol{\theta}}(\boldsymbol{x})}{Z_{\tilde{p}_{\boldsymbol{\theta}}}}$. After some manipulation, we have

$$
\sum_{\boldsymbol{x} \in \mathbb{B}^*}\frac{p^2_{\boldsymbol{\theta}}(\boldsymbol{x})}{q(\boldsymbol{x})} = \underbrace{\frac{Nk + Z_{\tilde{p}_{\boldsymbol{\theta}}}{}^2}{Z_{\tilde{p}_{\boldsymbol{\theta}}}{}^2}}_{=s \leq Nk+1}.
$$

Since $\forall \boldsymbol{x} \in \mathbb{B}^*$, $\frac{p^2_{\boldsymbol{\theta}}(\boldsymbol{x})}{q(\boldsymbol{x})} \geq 0$, we have

$$
\forall \boldsymbol{x} \in \mathbb{B}^*, \frac{p^2_{\boldsymbol{\theta}}(\boldsymbol{x})}{q(\boldsymbol{x})} \leq s; \tag{5}
$$

in particular, if $\boldsymbol{x}'$ encodes a halting trace of $M_b$, from §3.1 we know

$$
\begin{aligned}
p_{\boldsymbol{\theta}}(\boldsymbol{x}') &= \frac{\tilde{p}_{\boldsymbol{\theta}}(\boldsymbol{x}')}{Z_{\tilde{p}_{\boldsymbol{\theta}}}} \\
&= \frac{1 + \left(\frac{1}{3}\right)^{|\boldsymbol{x}'|+1}}{2} \\
&> 1/2.
\end{aligned}
\tag{6}
$$

Combining equations (5) and (6), we have for any halting trace $\boldsymbol{x}'$ of $M_b$,

$$
\frac{1}{4q(\boldsymbol{x}')} < \frac{p_{\boldsymbol{\theta}}^2(\boldsymbol{x}')}{q(\boldsymbol{x}')} \le s.
$$

After some more arrangement,

$$
q(\boldsymbol{x}') \ge \frac{1}{4s} \ge \frac{1}{4(Nk+1)} \ge \frac{1}{4(Nc+1)} > 0.
$$

As in the proof of Theorem 5, the existence of such a $q$ allows us to either prove or disprove the consistency of ZFC. For the sake of completeness, we include a proof sketch below:

1. By our assumptions, $q \in$ LN scores every string $\boldsymbol{x}$ as $q_{\boldsymbol{\theta}}(\boldsymbol{x})f(\boldsymbol{\theta}) \in \mathbb{Q}_{>0}$. Also, because $q$ is consistent and locally normalized, there exists $n \in \mathbb{N}$ such that we can prove $\forall \boldsymbol{x} \in \mathcal{X}_{>n}\{\boldsymbol{x} : \boldsymbol{x} \in \mathbb{B}^*, |\boldsymbol{x}| > n\}, q(\boldsymbol{x}) < \frac{1}{4(Nc+1)}$ (Proposition 1). We will just prove the existence of $n$ constructively by enumerating the strings in $\mathbb{B}^*$ in increasing length. Let the complement set $\mathcal{X}_{\le n} = \mathbb{B}^* - \mathcal{X}_{>n}$.
2. The proof then examines the finitely many strings in $\mathcal{X}_{\le n}$.
   (a) If any of these strings $\boldsymbol{x}'$ has $\tilde{p}_{\boldsymbol{\theta}}(\boldsymbol{x}') > 1$, then we know $\boldsymbol{x}'$ encodes an accepting trace of $M_b$ (§3.2). Therefore, $M_b$ halts, which implies there is a proof of the inconsistency of ZFC.
   (b) If none of these strings $\in \mathcal{X}_{\le n}$ has $\tilde{p}_{\boldsymbol{\theta}}(\boldsymbol{x}) > 1$, then we know there is also no string $\boldsymbol{x}'' \in \mathcal{X}_{>n}$, such that $\tilde{p}_{\boldsymbol{\theta}}(\boldsymbol{x}'') > 1$, since $\tilde{p}_{\boldsymbol{\theta}}(\boldsymbol{x}') > 1 \iff \tilde{b}(\boldsymbol{x}') > 1 \iff M_b$ halts; and we have already shown that for any accepting trace $\boldsymbol{x}'$ of $M_b$, $q(\boldsymbol{x}') \ge 1/4(Nc+1)$. Therefore, $M_b$ does not halt, which implies there is a proof of the consistency of ZFC.

Assuming ZFC is consistent, neither a proof of ZFC, or a disproof of ZFC, is possible. We have therefore therefore arrived at a contradiction. Therefore, such a $q$ does not exist. □

**Theorem 7.** *Let $\boldsymbol{\Theta}$ be an EC-complete family. There is no algorithm that takes $\boldsymbol{\theta}_1 \in \boldsymbol{\Theta}$, $\boldsymbol{\theta}_2 \in \boldsymbol{\Theta}$ as input, and decides whether $\tilde{p}_{\boldsymbol{\theta}_1}$ and $\tilde{p}_{\boldsymbol{\theta}_2}$ are the same weighted language.*

*Proof.* Assuming to the contrary that such an algorithm DISTINCT exists. We would be able to reduce HALT of input-free Turing machines to DISTINCT: we first construct $\boldsymbol{\theta}_1 \in \boldsymbol{\Theta}$ such that $\tilde{p}_{\boldsymbol{\theta}_1}(\boldsymbol{x}) = 1/3^{|\boldsymbol{x}|+1}$. We then construct $\boldsymbol{\theta}_2 \in \boldsymbol{\Theta}$ such that $\tilde{p}_{\boldsymbol{\theta}_2} = \tilde{p}_{M,1} \in \mathcal{G}_1$ (§3.1). DISTINCT$(\boldsymbol{\theta}_1, \boldsymbol{\theta}_2)$ is YES if and only if $M$ halts. □

**Theorem 8.** *For any $\tilde{p} \in$ EC and for any EC-complete family $\boldsymbol{\Theta}$, there is no algorithm BETTER$_{\tilde{p}}$ that takes two parameter vectors $\boldsymbol{\theta}_1 \in \boldsymbol{\Theta}$, $\boldsymbol{\theta}_2 \in \boldsymbol{\Theta}$, and returns YES if $\mathrm{KL}(p||p_{\boldsymbol{\theta}_1}) \ge \mathrm{KL}(p||p_{\boldsymbol{\theta}_2})$, and NO otherwise, where $p, p_{\boldsymbol{\theta}_1}, p_{\boldsymbol{\theta}_2}$ are string distributions defined by $\tilde{p}, \tilde{p}_{\boldsymbol{\theta}_1}, \tilde{p}_{\boldsymbol{\theta}_2}$ respectively.*

*Proof.* By contradiction; assume that for some $\tilde{p} \in$ EC, BETTER$_{\tilde{p}}$ exists. We know there exists a weighted Turing machine (also denoted as $\tilde{p}$) that on any string $\boldsymbol{x}$, terminates in $O(\mathrm{poly}(|\boldsymbol{x}|))$ and outputs $\tilde{p}(\boldsymbol{x})$ (Appendix B).

We show how we can reduce from HALT to BETTER$_{\tilde{p}}$. Given any arbitrary input-less Turing machine $M$, we can define a new weighted Turing machine $\tilde{p}'_M$ that on input string $\boldsymbol{x}$, $\tilde{p}'_M$ first simulates $\tilde{p}$ on it and keeps a record of $\tilde{p}(\boldsymbol{x})$ somewhere on the tape. $\tilde{p}'_M$ then verifies whether $\boldsymbol{x}$ is an encoded accepting trace of $M$. If $\boldsymbol{x}$ is indeed an encoded accepted trace of $M$, $\tilde{p}'_M$ outputs $\tilde{p}(\boldsymbol{x}) + 1$. Otherwise $\tilde{p}'_M$ outputs $\tilde{p}(\boldsymbol{x})$.

Let $\boldsymbol{\theta}_1 \in \boldsymbol{\Theta}$ parametrize $\tilde{p}'_M$, and let $\boldsymbol{\theta}_2 \in \boldsymbol{\Theta}$ parametrize $\tilde{p}$, such that $\tilde{p}_{\boldsymbol{\theta}_1}(\boldsymbol{x}) = \tilde{p}'_M(\boldsymbol{x}), \tilde{p}_{\boldsymbol{\theta}_2}(\boldsymbol{x}) = \tilde{p}(\boldsymbol{x}), \forall \boldsymbol{x} \in \mathbb{B}^*$.

We know that $\text{KL}(p||p) = 0$ for any distribution $p$. $\text{BETTER}_{\tilde{p}}(\theta_1, \theta_2)$ returns YES if and only if $\text{KL}(p||p_{\theta_1}) = 0 \geq \text{KL}(p||p_{\theta_2})$, which implies that $\tilde{p}_{\theta_2}$ and $\tilde{p}_{\theta_1}$ define the same distribution, which in turn implies that $\forall x \in \mathbb{B}^*, \tilde{p}(x) = \tilde{p}'_M(x)$, and that $M$ never halts. Similarly, $\text{BETTER}_{\tilde{p}}(\theta_1, \theta_2)$ returns NO if and only if $M$ halts. We have thus completed our reduction from $\text{HALT}$; and therefore algorithm $\text{BETTER}_{\tilde{p}}$ does not exist for all $\tilde{p} \in \text{EC}$. □

**Theorem 9.** *Let $p$ be any LN weighted language. Any $\tilde{p} \in \text{EC}$ where $\forall x \in V^*, \tilde{p}(x) \leq p(x)$ has a computable $Z_{\tilde{p}}$.*

*Proof.* We prove Theorem 9 by constructing an algorithm $\hat{Z}_{\tilde{p}} : \mathbb{Q}_{>0} \to \mathbb{Q}_{\geq 0}$ that approximates $Z_{\tilde{p}}$ within any desired positive rational error $\epsilon$: namely $|\hat{Z}_{\tilde{p}}(\epsilon) - Z_{\tilde{p}}| \leq \epsilon$.

Let $\hat{Z}_n = \sum_{\ell=0}^n \sum_{x:|x|=\ell} p(x)$. We first observe that $\lim_{n\to\infty} Z_n = 1$ by Proposition 1. In other words, $\lim_{n\to\infty} 1 - \hat{Z}_n = 0$, or equivalently, given any $\epsilon > 0$, there exists $n \in \mathbb{N}$ such that for all $n' \geq n, n' \in \mathbb{N}$, $(1 - \hat{Z}_{n'}) < \epsilon$.

Therefore, given any $\epsilon > 0$, $\exists n \in \mathbb{N}$ that divides $\mathbb{B}^*$ in two sets: $X_{\geq n} = \{x : x \in \mathbb{B}^*, |x| \geq n\}$, where $\sum_{x \in X_{\geq n}} p(x) < \epsilon$, and $X_{<n} = \mathbb{B}^* - X_{\geq n}$. We are guaranteed to find $n$ by enumerating all candidates from $\mathbb{N}$.

We can thus implement $\hat{Z}_{\tilde{p}}$ as the following program: given $\epsilon \in \mathbb{Q}_{>0}$, we first find the smallest $n \in \mathbb{N}$, and partition $\mathbb{B}^*$ into two sets $X_{<n}, X_{\geq n}$ as we described in the previous paragraph. We then return $\hat{Z}_{\tilde{p}}(\epsilon) = \sum_{x \in X_{<n}} \tilde{p}(x)$.

We argue that $\hat{Z}_{\tilde{p}}$ is a computable function. We first repeat that since $n \in \mathbb{N}$ exists, we will find $n$ in finite time, simply by enumerating. And since the set $X_{<n} \subset \mathbb{B}^*$ is finite, $\hat{Z}_{\tilde{p}}(\epsilon) = \sum_{x \in X_{<n}} \tilde{p}(x)$ can computed in finite time (under our assumption $\tilde{p} \in \text{EC}, \forall x \in \mathbb{B}^*, \tilde{p}(x)$ can be computed in time $O(\text{poly}(|x|))$). Since the program we described above terminates in finite time, $\hat{Z}_{\tilde{p}}$ is a computable function.

What remains is to show that the approximation error $|Z_{\tilde{p}} - \hat{Z}_{\tilde{p}}(\epsilon)|$ is no greater than $\epsilon$; *i.e.*, that $|Z_{\tilde{p}} - \hat{Z}_{\tilde{p}}(\epsilon)| \leq \epsilon$.

It is easy to show that $0 \leq Z_{\tilde{p}} - \hat{Z}_{\tilde{p}}(\epsilon)$, after which we only need to show that $Z_{\tilde{p}} - \hat{Z}_{\tilde{p}}(\epsilon) \leq \epsilon$. Expressing both terms as sums, we have that $\sum_{x \in \mathbb{B}^*} \tilde{p}(x) - \sum_{x \in X_{<n}} \tilde{p}(x) = \sum_{x \in X_{\geq n}} \tilde{p}(x)$, which is a sum of nonnegative terms and thus nonnegative.

To show that $Z_{\tilde{p}} - \hat{Z}_{\tilde{p}}(\epsilon) \leq \epsilon$, we express both terms as sums again:

$$\sum_{x \in \mathbb{B}^*} \tilde{p}(x) - \sum_{x \in X_{<n}} \tilde{p}(x) = \sum_{x \in X_{\geq n}} \tilde{p}(x).$$

By the definition of $\tilde{p}$, we have $\sum_{x \in X_{\geq n}} \tilde{p}(x) \leq \sum_{x \in X_{\geq n}} p(x)$. The right-hand side is equal to $1 - \hat{Z}_n$, which by construction is less than $\epsilon$. Therefore, by substituting to get $Z_{\tilde{p}} - \hat{Z}_{\tilde{p}}(\epsilon) \leq 1 - \hat{Z}_n$ and using the transitive property of inequality, we have that $Z_{\tilde{p}} - \hat{Z}_{\tilde{p}}(\epsilon) < \epsilon$. Combined with the previous paragraph's result, we have shown that $|Z_{\tilde{p}} - \hat{Z}_{\tilde{p}}(\epsilon)| \leq \epsilon$.

□

# E  CONNECTION BETWEEN EXPRESSIVE SEQUENCE DISTRIBUTIONS AND LIN ET AL. (2021)'S PROOF THAT EC ≠ ELN

Our construction of expressive sequence distributions is similar to Lin et al. (2021)'s construction of an EC weighted language that is not in ELN. Whereas each weighted machine $\tilde{p}_{M,k} \in \mathcal{G}_k$ corresponds to execution traces of a deterministic Turing machine $M$, Lin et al. (2021) defined a single weighted language $\tilde{p}_{\text{ELN}}$ for all Turing machines. Fore the sake of completeness, we describe the definition of $\tilde{p}_{\text{ELN}}$ from Lin et al. (2021) below. Let enc be a prefix-free encoding function that maps a deterministic Turing machine $M$ to a Boolean string where the function inverse $\text{enc}^{-1}$ exists. And let $\tilde{p}_{\text{ELN}}$ be a weighted language over Boolean strings, where $\tilde{p}_{\text{ELN}}(x) = 1/3^{|x|+1}$ if $x$ is of the form $x^{(1)}x^{(2)}$, where $x^{(1)}$ encodes some Turing machine $M$, and $x^{(2)}$ encodes an accepting execution path of $M$ on an empty input. (Such a path may be

represented as a sequence of transitions of $M$ that begins with an initial state and ends at an accepting state.) Lin et al. (2021) then showed that prefix probabilities of $\tilde{p}_{\text{ELN}}$ are uncomputable under an ELN, since HALT reduces to computing $Z(\boldsymbol{x}^{(1)})$.

Theorem 9 implies that $\tilde{p}_{\text{ELN}}$ we have just described above has a computable partition function:

**Theorem 10.** *Let $Z$ be the partition function of $\tilde{p}_{\text{ELN}}$. $Z$ is computable.*

*Proof.* Let $p(\boldsymbol{x}) = 1/3^{|x|+1}$. $p \in \text{ELN} \subset \text{LN}$ because $p(\cdot \mid \hat{\boldsymbol{x}}) = 1/3$ for all valid prefixes $\hat{\boldsymbol{x}}$. Since $\tilde{p}_{\text{ELN}} \in \text{EC}$ by Lin et al. (2021, Theorem 5) and $\forall \boldsymbol{x} \in \mathbb{B}^*, \tilde{p}_{\text{ELN}}(\boldsymbol{x}) \leq p(\boldsymbol{x})$, we have $Z$ is computable by Theorem 9. $\qquad\square$

Similarly, the 'sparse version' of $\tilde{p}_{\text{ELN}}$ introduced in (Lin et al., 2021, Theorem 6) can be shown to have a computable partition function as well.

Since one of our goals is to clearly demonstrate that we can construct a single weighted language $\in \text{EC}$ that has an uncomputable partition function assuming ZFC (*e.g.*, $\tilde{b}$ in §3.2), we define weighted languages in $\mathcal{G}_k$ to have at most one 'high' weight, as opposed to $\tilde{p}$ in Lin et al. (2021, Theorem 5), where each $\boldsymbol{x}^{(1)}$ that encodes a halting machine has a 'high weight' suffix $\boldsymbol{x}^{(2)}$.

In fact, under our construction we can directly show $\text{EC} \neq \text{ELN}$, using the uncomputability of $\tilde{b}$ from Theorem 3:

**Corollary 1** (EC $\neq$ ELN under ZFC; slightly weaker version of Theorem 5 in Lin et al. (2021)). *Assuming the axiomatic system of ZFC, $\text{EC} \neq \text{ELN}$.*

*Proof.* We know there exists a weighted language $\tilde{b} \in \text{EC}$ (§3.2) that has an uncomputable partition function. Since $\tilde{b} \in \text{EC}$, $\tilde{b}(\boldsymbol{x}) \in \mathbb{Q}_{\geq 0}, \forall \boldsymbol{x} \in \mathbb{B}^*$. Therefore for all strings $\boldsymbol{x} \in \mathbb{B}^*$, $b(\boldsymbol{x}) = \tilde{b}(\boldsymbol{x})/Z_{\tilde{b}}$ is an uncomputable number.

Assuming to the contrary that $\tilde{b} \in \text{ELN}$. By definition $b(\boldsymbol{x}) = \prod b(x_t \mid \boldsymbol{x}_{<t})$, where each $b(x_t \mid \boldsymbol{x}_{<t}) \in \mathbb{Q}_{\geq \nu}$ and is computable. Since the set of computable numbers is closed under multiplication, $b(\boldsymbol{x})$ would also be computable, which contradicts our assumption. Therefore $\tilde{b} \neq \text{ELN}$, which implies $\text{EC} \neq \text{ELN}$. $\quad\square$

## F  NEGATIVE RESULTS POSSIBLE ON FINITE PARAMETER SUBSPACES

In §3.2, we mention that the pathological EBM $\tilde{b}$ can show negative results regarding finite subsets of $\boldsymbol{\Theta}$. Here, we provide a concrete example.

**Corollary 2.** *Assuming ZFC axioms and assuming they are consistent, there exists $\boldsymbol{\theta} \in \boldsymbol{\Theta}$ such that $Z_{\tilde{p}_{\boldsymbol{\theta}}} \in \mathbb{Q}_{>0}$ exists, but there is no algorithm $\hat{Z}_{\text{proof}}$ that takes $\boldsymbol{\theta} \in \boldsymbol{\Theta}$ as input, and outputs a set of strings $\{\boldsymbol{x}_n : n < N\}, N \in \mathbb{N}$ where*

- *there is a proof that $\sum_{n=1}^N \tilde{p}_{\boldsymbol{\theta}}(\boldsymbol{x}) \geq 1/2 Z_{\tilde{p}_{\boldsymbol{\theta}}}$; or*
- *there is a proof that $\sum_{n=1}^N \tilde{p}_{\boldsymbol{\theta}}(\boldsymbol{x}) < 1/2 Z_{\tilde{p}_{\boldsymbol{\theta}}}$.*

*Proof.* Assuming to the contrary that $\hat{Z}_{\text{proof}}$ existed. We construct $\boldsymbol{\theta} \in \boldsymbol{\Theta}$ such that $\forall \boldsymbol{x} \in \mathbb{B}^*, \tilde{p}_{\boldsymbol{\theta}}(\boldsymbol{x}) = \tilde{b}(\boldsymbol{x})$. Either proof resulting from $\hat{Z}_{\text{proof}}(\boldsymbol{\theta})$: $\{\boldsymbol{x}_n : n < N\}$ can be used to prove or disprove the consistency of ZFC. $\qquad\square$

Corollary 2 states that there exists a 'problematic EBM' — namely $\tilde{b}$ — where we cannot guarantee to well approximate its partition function, by accumulating finitely many string weights, regardless of the manner of accumulation (*i.e.*, how we choose strings) or the number of strings we enumerate over. We discuss this in further details in §2.5.

## G  IMPOSSIBILITY OF MODEL SELECTION IN FIXED-SIZE TRANSFORMER EBM FAMILIES

Theorem 11 is a sibling theorem to Theorem 8. Just as we show $Z_{\tilde{b}}$ is uncomputable (§3.2), we can prove that model selection is not only impossible for EC-complete parametric families (where the length of a parameter vector is unbounded), but also impossible for fixed-size Transformer EBMs with a large enough

embedding size.[15]

**Theorem 11.** *Assuming ZFC as our axiomatic system, for any $\tilde{p} \in EC$, there exists $d_0 \in \mathbb{N}$ such that for all $d \geq d_0$, $\tilde{p}$ can be captured by one-encoder-layer four-decoder-layer Transformer networks $\boldsymbol{\Theta}^{(d)}$ (Theorem 1) with embedding size $d$, where there is no provably correct algorithm $\textsc{Better}_{\tilde{p},d}$ that takes two parameter vectors $\boldsymbol{\theta}_1 \in \boldsymbol{\Theta}^{(d)}$, $\boldsymbol{\theta}_2 \in \boldsymbol{\Theta}^{(d)}$, and returns YES if $\mathrm{KL}(p||p_{\boldsymbol{\theta}_1}) \geq \mathrm{KL}(p||p_{\boldsymbol{\theta}_2})$, and NO otherwise, where $p, p_{\boldsymbol{\theta}_1}, p_{\boldsymbol{\theta}_2}$ are string distributions defined by $\tilde{p}, \tilde{p}_{\boldsymbol{\theta}_1}, \tilde{p}_{\boldsymbol{\theta}_2}$ respectively.*

*Proof.* Let $M_b$ be the input-free unweighted Turing machine we built in our proof of Theorem 3, whose behavior is independent of ZFC. Let $M_0$ be any weighted Turing machine that defines the EC weighted language $\tilde{p}$. And let $M_1$ be a weighted Turing machine that weights $\boldsymbol{x}$ as $\tilde{p}(\boldsymbol{x}) + 1$ if and only if $\boldsymbol{x}$ encodes an accepting trace of $M_b$; and $M_1$ weights $\boldsymbol{x}$ as $\tilde{p}(\boldsymbol{x})$ otherwise. Since checking whether $\boldsymbol{x}$ is a valid trace of $M_b$ is in $O(\mathrm{poly}(|\boldsymbol{x}|))$, $M_1$ defines an EC weighted language.

Let $p_{M_1}$ be the string distribution defined by $M_1$. By an argument similar to our proof of Theorem 8, no algorithm that is provably correct can decide if $\mathrm{KL}(p||p_{M_1}) \geq \mathrm{KL}(p||p)$.

We note that for any weighted Turing machine $M$ with fewer than $n$ states, we can build another weighted Turing machine $M'$ which has $n$ states, such that $M$ and $M'$ define the same weighted language, simply by having (finitely many) additional unreachable states in $M'$. Since any weighted Turing machine with $n$ states can be implemented as a 1-encoder-layer-4-decoder-layer Transformer networks with an embedding size $\in O(n)$, it follows that there exists $d_0 \in \mathbb{N}$ such that both $M_0$ and $M_1$ can be encoded as parameter vectors within a fixed-size model family with $d \geq d_0$. $\qquad\square$

---

[15]We use $\boldsymbol{\Theta}^{(d)}$ to denote a Transformer family with embedding size $d$.

