# OpenReview forum: "On the Uncomputability of Partition Functions in Energy-Based Sequence Models"
_ICLR.cc/2022/Conference — ICLR 2022 Spotlight_

### Official Review · Reviewer_EUof · 2021-10-21

**Correctness:** 4
**Technical Novelty And Significance:** 4
**Empirical Novelty And Significance:** 1
**Recommendation:** 8
**Confidence:** 3

**Main Review:**

This paper tackles the important question of what we loose in theoretical guarantees if we remove all restrictions on the energy-based model. As such, this paper provides an important answer to conjectures raised in (Lin et al. 2021).
The paper is technically sound and well written to the degree that the arguments are also comprehensible for non-experts; I honestly enjoyed reading the paper.
One might argue that the findings are somewhat abstract as they only hold for energy-based models without any restrictions on the model expressively. The theoretical properties (e.g., regarding randomized algorithms) are intriguing at first, but less so considering that we implicitly assume that $\tilde{p}$ has infinite support. Granted, the authors are quite open about the -- lack of -- assumptions and discuss the implications of typical model restrictions in depth in Sec. 7. Nonetheless, I would have preferred a brief discussion about the practical relevance of EC-complete parametric families in Sec. 2.4. (and in the introduction) and to what degree certain model restrictions lead to non-complete families. That is if the authors would state the scope of the current work more blatantly.

I have a couple of further remarks and questions:
* I cannot follow the logic from Prop. 1 that infinite-length sequences have zero mass; that is, I do not understand why $\epsilon$ must go to zero for $n \rightarrow \infty$.
* In Sec. 4 it is stated that no estimate exists with a bias within a multiplicative factor. If I understand Lm. 1 correctly, such an estimate does exist, however; although only with probability < 1/3. Can you explain this discrepancy?
* Most arguments hinge on the undecidability of the HALT problem. I think that it would help the reader if that was stated explicitly in the text.
* Thm. 5 states that rejection sampling does not work for EC-complete families. I get the argument, but I was left wondering whether it might be possible to quantify the approximation error one introduces when relying on randomized methods nonetheless.

**Summary Of The Paper:**

The paper studies the trade-off between expressiveness and computability (of the partition function) for energy-based sequence models. The theoretical results show that the high expressively of unrestricted energy based models comes at the cost of un-computability and in-approximability of the partition function. These negative results further show that rejection and importance sampling are not a panacea either.

**Summary Of The Review:**

The paper considers an important problem, is technically sound (at least from what I can tell), and well-written. Although the practical implications are not immediately clear, I am convinced that it is actually relevant to understand the theoretical limitation of considering energy-based models with ever-increasing expressiveness.

---

> ### Author Response · Authors · 2021-11-14
> **Response to Reviewer EUof**
>
> We thank Reviewer EUof for their valuable feedback! We are also grateful that they like our manuscript.
>
> > I would have preferred a brief discussion about the practical relevance of EC-complete parametric families in Sec. 2.4. (and in the introduction) and to what degree certain model restrictions lead to non-complete families. That is if the authors would state the scope of the current work more blatantly.
>
> We emphasize that our negative results apply to model families that may not be EC-complete as well (e.g. fixed-size Transformer EBMs), if they are capable enough to parametrize certain EC weighted languages whose partition function is known to be uncomputable (e.g. the example we give in Sec 3.2). We have revised the manuscript to make this more explicit (Sec 1 and start of Sec 7; also formally stated in Theorem 11). Some discussions on model restrictions that make sequence models non-EC-complete are in Sec 7 as well. We will continue to revise the manuscript for a better presentation. We thank Reviewer EUof for this helpful suggestion!
>
> > I cannot follow the logic from Prop. 1 that infinite-length sequences have zero mass; that is, I do not understand why $\epsilon$ must go to zero for $n\rightarrow \infty$.
>
> Assuming to the contrary that $\epsilon$ is bounded away from zero as $n$ goes to $\infty$, the sum of all string weights would diverge, which contradicts our assumption that the partition function exists.
>
> > In Sec. 4 it is stated that no estimate exists with a bias within a multiplicative factor. If I understand Lm. 1 correctly, such an estimate does exist, however; although only with probability < 1/3. Can you explain this discrepancy?
>
> Yes. Speaking loosely, Lemma 1 states that there is no randomized estimate that is usually accurate (with probability > ⅔). But Lemma 1 says nothing about the mean -- it did not rule out the possibility that there might be an estimator that has huge variance, but is accurate on average (i.e. sample mean asymptotically approaches the true partition function). Thm 4 says this is impossible, too.
>
> > Most arguments hinge on the undecidability of the HALT problem. I think that it would help the reader if that was stated explicitly in the text.
>
> We have revised the manuscript to add more background on decidability and HALT as Appendix G. (We wanted to avoid massive renumbering that would make reviewer comments stale, so most new appendices were added at the end for now.) We thank Reviewer EUof for this suggestion.
>
> > Thm. 5 states that rejection sampling does not work for EC-complete families. I get the argument, but I was left wondering whether it might be possible to quantify the approximation error one introduces when relying on randomized methods nonetheless.
>
> Theorem 5 states that there are EBMs where we can never prove whether rejection sampling is possible under any proposal distribution ($q$)/constant ($c_q$) combination. Since we cannot even prove that a proposal-constant combination is valid in the first place, and any violation (that is, $\frac{\tilde{p}(\mathbf{x})}{q(\mathbf{x})} > c_q$) can be arbitrarily large, we do not think it is possible to quantify the approximation error introduced by having an inappropriate proposal distribution/constant combination.

---

> > ### Comment · Reviewer_EUof · 2021-11-25
> > **Thanks for the clarifications**
> >
> > I have read the response of the authors. The authors have answered all my questions in detail and clarified certain aspects that were not that obvious to me originally.
> > After also reading the other reviews, I will stick to my decision that the paper makes a solid contribution and keep my score.

---

### Official Review · Reviewer_K6DS · 2021-11-02

**Correctness:** 4
**Technical Novelty And Significance:** 2
**Empirical Novelty And Significance:** Not applicable
**Recommendation:** 6
**Confidence:** 3

**Main Review:**

The main strength of this submission is its cohesive conceptual message. This work studies the viability of EBMs from an unconventional angle: whereas one's initial reaction to the complexity of computing the partition function for an EBM might be to point out that it is obviously computationally expensive, the findings in this paper show that in fact this task isn't even computable. A related strength is that the paper systematically pursues this idea from a variety of perspectives, e.g. exact/approximate/asymptotic + deterministic/randomized estimation, failure of off-the-shelf techniques like rejection/importance sampling, related tasks like identifiability and model selection.

As for weaknesses, the actual technical content isn't particularly involved and arguably not that surprising given that families like Transformers are known to be Turing-complete, though perhaps this is an unfair criticism given that the contribution of this work is primarily conceptual. Regarding the writing, I did feel that the important preliminaries were introduced in a rushed fashion; for instance, even though they figure in a majority of the theorems in this submission, their definition of "EC-complete parametric families" was extremely difficult to parse, in part because their definition of seq-to-seq models was similarly difficult to understand. That said, the formalism doesn't figure that heavily in the proofs as the arguments are ultimately primarily about Turing machines anyways.

**Summary Of The Paper:**

This paper studies "efficiently computable (EC) energy-based sequence models (EBMs)" which are simply sets of strings equipped with nonnegative weights that can be computed by a poly-time Turing machine (upon normalization, the weights induce a probability distribution over the strings). In practice, it is common for these weights to be computed by neural networks. In fact, certain neural sequence model families like RNNs and Transformers are expressive enough to be Turing-complete. In this paper, the authors find that the expressivity of such sequence model families, so-called "EC-complete parametric families", comes at a significant cost in terms of *computability/decidability* of various primitives for inference. For instance, they show that if one could actually take in any vector of parameters specifying a model in such a family and output the corresponding partition function (sum of the weights of all strings) even approximately in expectation, then would be able to decide the halting problem (Theorem 2, 4). They also exhibit a single efficiently computable EBM for which proving (within ZFC) that the partition function is one of two possible values would either disprove Godel's second incompleteness theorem or disprove consistency of ZFC (Theorem 3); this can be extended to show that *asymptotic* estimates like rejection sampling and importance sampling will also fail (Theorems 5, 6). Along similar lines, they can reduce from the halting problem to other tasks like deciding whether two given parameter vectors give rise to the same EBM ("parameter identifiability", Theorem 7), or deciding which of two given parameter vectors gives rise to an EBM which is distributionally closer to some other given EBM ("model selection", Theorem 8).

In terms of techniques, the reductions from the halting problem are all based on a simple weight function that is tiny unless if the string corresponds to a valid accepting trace of an input-free Turing machine M. The point is that the weight for any string x can be computed by M simply by simulating M on x, but the partition function is large iff M halts. The ZFC results follow by taking M which enumerates all provable propositions under ZF and halts iff it proves 1 = 0.

**Summary Of The Review:**

I am not an expert in this area so I'm not sure to what extent the field was already aware of issues of computability. Superficially the connection between EC and the halting problem does seem quite straightforward, but if this was not observed or studied in any depth previously in the EBMs literature, I think the submission is a worthy conceptual contribution that should be accepted.

---

> ### Author Response · Authors · 2021-11-14
> **Response to Reviewer K6DS**
>
> We thank Reviewer K6DS for their time, and also for their detailed review!
>
> > I did feel that the important preliminaries were introduced in a rushed fashion; for instance, even though they figure in a majority of the theorems in this submission, their definition of "EC-complete parametric families" was extremely difficult to parse, in part because their definition of seq-to-seq models was similarly difficult to understand.
>
> We have revised Sec 2.4 to make it more approachable (and will continue to refine our writing even after the rebuttal period ends). We thank Reviewer K6DS for this comment.

---

> > ### Comment · Reviewer_K6DS · 2021-11-24
> > **Thanks for the revisions!**
> >
> > Apologies for the delayed response, this is to acknowledge I have read the rebuttal and the other reviews. I continue to think the submission is a solid contribution that merits acceptance and will keep my score as before.

---

### Official Review · Reviewer_NAcm · 2021-11-03

**Correctness:** 3
**Technical Novelty And Significance:** 3
**Empirical Novelty And Significance:** 2
**Recommendation:** 8
**Confidence:** 4

**Main Review:**

**Strengths:**

- This paper rigorously explores the computational barriers of calculating the partition function and model selection, both of which are fundamental machine learning tasks

- A specific EBM is exhibited whose partition function is uncomputable (even unprovable) assuming the consistency of ZFC -- to my knowledge this is a new connection between logic and machine learning.

- Sections 7 and 9 point to more interesting work to be done

**Concerns:**

- The uncomputability of the partition function is not necessarily a barrier to using EBMs in practice if the task at hand does not require sampling or model selection

- The uncomputability of the partition function seems to be known. See the "Remark on computability" in Lin et al '21. Also the proof of Lin et al's Theorem 5 uses a very similar construction to the paper under the review, and this should be mentioned.

- As this paper admits, the EBMs studied are pathological and not necessarily an obstruction to practice. There is also a lack of discussion on polynomial-time computation of partition functions in less pathological situations (see [1,2,3] below)

- Only rather specific asymptotic estimates of the partition function are ruled out. Perhaps this deserves further study.

**Comments**

- pg 4: I had trouble understanding the phrase "many estimation procedures are anytime algorithms". Can you explain this a bit more? It seemed that asymptotic estimates defined here capture the idea of the sample size going to infinity, but I am unclear on the details.

- pg 6, Theorem 4: Is it possible to simplify the proof of Theorem 4 as follows? Given a randomized estimate $\hat Z_\epsilon$, one can compute $\mathbb{E}[\hat Z_\epsilon]$ by derandomization. If $\varepsilon^{-1} Z \leq \mathbb{E}[\hat Z_\epsilon] \leq \varepsilon Z$ holds, then the exact estimate $\mathbb{E}[\hat Z_\epsilon]$ satisfies Equation (2) from Lemma 1 with probability $1$. But this is impossible by Lemma 1.

- pg 7, Theorem 7: In most applications, it is not necessary to verify identifiability with an algorithm. Perhaps it is worth only mentioning the model selection result in the main text and using the identifiability as a proof tool in the Appendix.

- pg 9, Conclusion: Lin et al '21 are aware of computability issues arising from EBMs. To my understanding, the work under review continues the conversation about uncomputability and says some interesting new things about it. And again, just because the partition function is uncomputable, it doesn't mean that EBMs are useless for all applications. I think the conclusion should be rephrased to acknowledge these points.

- pg 20: Table C.1 is interesting and could be cited in the main text.

**Minor issues**

- pg2, Table 1: This table is hard to understand without knowing the definitions. The phrase "suspected to be useless in general" is too informal.

- pg 5, Theorem 3: Please describe what $\tilde b$ is around here.

- pg 5: "The weighted language... by simulating $x$ on $M$." Please describe this more  formally. Lin et al's proof of Theorem 5 uses convenient notation for this.

- pg 5: "We discuss this in further detail in Section 2.5" This is a backward reference; is it a typo?

- pg 6: A brief definition of rejection sampling would be helpful at the beginning of Section 5.1.

- pg 8: $\Psi_{\vert x \vert}$ is not defined when describing FGGs.

- pg 19: "for any halting trace $x'$ of $M_b$" seems to be a typo.

**References**

[1] M. Luby and E. Vigoda, “Fast convergence of the glauber dynamics for sampling independent sets,” Random Structures and Algorithms, vol. 15, no. 3-4, pp. 229–241, 1999.

[2] A. Galanis, D. Stefankovic, and E. Vigoda, “Inapproximability of the partition function for the antiferromagnetic Ising and hard-core models,” arXiv preprint arXiv:1203.2226, 2012.

[3] M. Jerrum and A. Sinclair, “Polynomial-time approximation algorithms for the Ising model,”
SIAM Journal on computing, vol. 22, no. 5, pp. 1087–1116, 1993

**Summary Of The Paper:**

This theoretical work highlights uncomputability issues arising with energy-based sequence models. It is shown that an EC-complete family (a certain computational model capturing neural networks and transformers that is essentially equivalent to weighted Turing machines) cannot approximate the partition function of energy-based sequence models (EBMs) even given an unlimited amount of time and space. A consequence is the impossibility of model selection for EBMs. This paper also rules out popular estimators such as rejection and important sampling. This work concludes with a discussion of restricted EBMs that avoid such uncomputability issues.

**Summary Of The Review:**

This paper develops an interesting point of view on machine learning and computability. The proofs seem to be correct. I support accepting this paper once it is revised to take into account concerns raised above.

---

> ### Author Response · Authors · 2021-11-14
> **Response to Reviewer NAcm (1/2)**
>
> This review was incredibly thorough and showed a deep engagement with our work; for this, we thank reviewer NAcm.
>
> > The uncomputability of the partition function is not necessarily a barrier to using EBMs in practice if the task at hand does not require sampling or model selection
>
> Indeed, some tasks (e.g. evaluating unnormalized string scores) do not require the computation of the partition function, so its uncomputability might not concern us. We have revised our manuscript to discuss the scope of our work more explicitly in footnote 2, and Sec 9. We have also revised Sec 9 to discuss future directions.
>
> > The uncomputability of the partition function seems to be known. See the "Remark on computability" in Lin et al '21.
>
> Again, we thank Reviewer NAcm for this comment, and have revised our manuscript (Sec 8, Appendix E) to include more discussion on Lin et al. ‘21. We note that the technical content and purpose of ‘remark on computability’ in Lin et al. ‘21 differ rather significantly from our work. While they did raise the point that a weighted language’s partition function may be uncomputable, their only reason appears to be that an infinite sum of string weights can be uncomputable. They did not provide any specific example, much less a constructive proof. Moreover, none of the weighted languages constructed in Lin et al. ‘21 has an uncomputable partition function: while in the proof of their Theorem 5 they constructed a weighted language $\tilde{p}$ that has uncomputable autoregressive factors, this weighted language actually has a computable partition function (as we show in Theorem 10). Same for the weighted language they constructed in their Theorem 6. We have revised our manuscript to include simple proofs that those weighted languages have computable partition functions (Theorem 10 in Appendix E). Instead, they emphasized that under their settings, thanks to the oracular access to trained parameter strings, arbitrarily good approximations of the (possibly uncomputable) partition function can be memorized in the (autoregressive) model parameters.
>
> As Reviewer NAcm astutely notes, our work can be seen as continuing a conversation on uncomputability started by Lin et al. We think there is an interesting contrast between the stances of Lin et al. 21 and ours: while Lin et al. did note the (un)computability problem of $Z$, they saw this as a trivial issue from the model capacity viewpoint, since good approximations can be stored as parameters (as long as we can find them). However in our work we regard the uncomputability problem as a parameter estimation disaster -- there will be no guarantee of good approximations under standard computation models, which breaks several other things.
>
> > Also the proof of Lin et al's Theorem 5 uses a very similar construction to the paper under the review, and this should be mentioned.
>
> Our construction of expressive sequence distributions ($\mathcal{G}_k$) is indeed inspired by the weighted language used in Lin et al’s Thm 5 proof, with a modification to ensure each weighted language has at most one ‘high weight’ string. In fact, under our construction, Thm 5 of Lin et al. follows as a corollary after our Thm 3 (under the assumption of ZFC). We have revised our manuscript (Appendix E) to acknowledge the connection between two constructions, and added the new (admittedly simple) proof.
>
> > As this paper admits, the EBMs studied are pathological and not necessarily an obstruction to practice.
>
> While the pathological EBMs we study may seem toy-ish to ML practitioners, we argue that a much bigger problem is that their existence marks a sort of expressiveness upper bound of learnable EBMs over sequences -- once the expressiveness of an EBM model family surpasses this mark (and can parametrize these pathological EBMs), then model selection is no longer possible for this family. Specifically, we can use such results to show model selection is impossible for large enough fixed-size Transformer EBMs (Theorem 11). Our only realistic choice is to cripple model expressiveness to the degree that it provably cannot contain those pathological models (a la Sec 7). We have revised our manuscript (first paragraph of Sec 7 and Appendix F) to make this point more explicit.

---

> ### Author Response · Authors · 2021-11-14
> **Response to Reviewer NAcm (2/2)**
>
> > There is also a lack of discussion on polynomial-time computation of partition functions in less pathological situations (see [1,2,3] below)
>
> We thank the reviewer for suggesting these references. Our paper would certainly benefit from a discussion of the complexity aspects of partition function approximation, when we 1) limit the maximum string length and 2) limit the amount of interaction symbols can have among each other. Indeed there are settings where the partition function can be approximated by FPTAS/FPRAS algorithms [1,3]; but when we relax their interaction constraint by a bit then the partition function becomes likely inapproximable [2]. Another important related work would be [Chandrasekaran et al. (2008)](https://dl.acm.org/doi/10.5555/3023476.3023485), which stated the best worst-case estimation of a graphical model’s partition function cannot have runtime polynomial in the graph’s treewidth. We have revised our manuscript to incorporate the discussion above (Sec 7).
>
> > Only rather specific asymptotic estimates of the partition function are ruled out. Perhaps this deserves further study.
>
> We agree that the class of all asymptotic estimates as a whole requires further study. We revised our manuscript to indicate that we leave this as future work (Sec 5).
>
> > I had trouble understanding the phrase "many estimation procedures are anytime algorithms". Can you explain this a bit more? It seemed that asymptotic estimates defined here capture the idea of the sample size going to infinity, but I am unclear on the details.
>
> An asymptotic estimate can be deterministic under our definition: for example, $\hat{Z}_{\textrm{asym}}$ at the start of Sec 5. We have edited Sec 2.5 to add a forward reference.
>
> > pg 6, Theorem 4: Is it possible to simplify the proof of Theorem 4 as follows? Given a randomized estimate $\mathbf{\hat{Z}_\epsilon}$, one can compute $\mathbb{E}[\mathbf{\hat{Z}_\epsilon}]$ by derandomization. If $\epsilon^{-1}Z \leq \mathbb{E}[\mathbf{\hat{Z}_\epsilon}] \leq \epsilon Z$ holds, then the exact estimate $\mathbf{\hat{Z}_\epsilon}$ satisfies Equation (2) from Lemma 1 with probability 1. But this is impossible by Lemma 1.
>
> We are very grateful for this suggestion from Reviewer NAcm. Indeed we can derandomize the expectation of the randomized estimate $\mathbf{\hat{Z}_\epsilon}$ directly, and do not need to rely on the uniform convergence of a sample mean to the normal distribution. Our original proof was inspired by the common practice to ‘derandomize’ a random variable by drawing multiple samples, and rely on the CLT to establish a confidence interval of the true mean. But for our Thm 4, we can directly derandomize the expectation -- that is, the proof of Theorem 4 directly follows our Lemma 2. Again we thank Reviewer NAcm! We have revised our manuscript (Thm 4) to incorporate this simplification.
>
> > pg 7, Theorem 7: In most applications, it is not necessary to verify identifiability with an algorithm. Perhaps it is worth only mentioning the model selection result in the main text and using the identifiability as a proof tool in the Appendix.
>
> We agree that model identifiability is not a crucial requirement in most applications. To keep theorem numbering identical between our original submission and subsequent revisions, we will move the current Thm 7 to the appendices in camera ready, if this paper is accepted.
>
> > pg 9, Conclusion: Lin et al '21 are aware of computability issues arising from EBMs. To my understanding, the work under review continues the conversation about uncomputability and says some interesting new things about it.
>
> We agree that this work is continuing a conversation about uncomputability, started by Lin et al. ‘21. We have revised our manuscript accordingly; see Appendix E.
>
> > And again, just because the partition function is uncomputable, it doesn't mean that EBMs are useless for all applications. I think the conclusion should be rephrased to acknowledge these points.
>
> We also agree that EBMs can still be immensely useful in many applications. A main message of this paper is pointing out certain uncomputability problems can arise during parameter estimation if the EBM is too powerful. But 1) EBM parameters are not necessarily learned from data and 2) depending on the task, it may be reasonable to cripple the model expressiveness (Sec 7). We have revised the manuscript to make these two points more clear. We have also revised Sec 9 to emphasize that we did not mean to invalidate the usefulness of EBMs.
>
> > pg 20: Table C.1 is interesting and could be cited in the main text.
>
> We agree. We have revised our manuscript to include a forward reference.
>
> We have also addressed other issues raised by Reviewer NAcm as 'Minor Issues'. We thank Reviewer NAcm again for their careful review!

---

> > ### Comment · Reviewer_NAcm · 2021-11-15
> > **Re:**
> >
> > Thanks to the authors for this careful, detailed reply and for responding to my concerns.

---

### Official Review · Reviewer_EwJC · 2021-11-05

**Correctness:** 4
**Technical Novelty And Significance:** 3
**Empirical Novelty And Significance:** Not applicable
**Recommendation:** 6
**Confidence:** 3

**Main Review:**

The paper is generally well written with a good exposition. One glaring exception (for me) is the 2nd paragraph of Section 2.4 where the reduction from neural sequence to formal languages is presented. This is crucial for understanding the paper, yet it is done quickly and I found it hard to follow. It may also be a good place to draw clear distinction from the work of Perez et al (2021). This (and Theorem 1) are key, after all, theorem 2 (for instance) is quite straightforward once the setup is established.

It would be nice to connect the 'recommendations' of Section 7 to actual EBM's used in practice.

**Summary Of The Paper:**

The paper works in a computational model where various architectures of energy-based-models are viewed as computational models for accepting languages. EBM's assign a weight to each string. The sum of weights is called the partition function and this needs to be computed (or at best approximated) if the energy is to be thought of as a probability measure. The paper shows that expressive EBM's are turning complete, and uses that to show that the partition function may be uncomputable. And that is even in the case the energy of a sequence could be computed in poly-time. The paper then shows some corollaries and variations of this result, showing that model selection is undecidable as well. The paper concludes by suggesting scenarios where the partition function is computable but this naturally comes at the expense of the expressivity of the model.

**Summary Of The Review:**

The paper is clear and interesting, but suffers from two major weaknesses:
1. It seems to me the big 'insight' is Turing completeness of the model, but this is (mostly) done in previous papers. Once this is established the remaining reductions are much less surprising.
2. I'm not sure the paper is a good fit for ICLR. It is mainly a complexity/computability paper where the models are related to ML. I'm skeptical that its insights are interesting for most of the ICLR audience  (If I'm wrong I'm  happy to be corrected).

---

> ### Author Response · Authors · 2021-11-14
> **Response to Reviewer EwJC**
>
> We’re grateful for your comments, and for finding the paper clear and interesting!
>
> > One glaring exception (for me) is the 2nd paragraph of Section 2.4 where the reduction from neural sequence to formal languages is presented. This is crucial for understanding the paper, yet it is done quickly and I found it hard to follow.
>
> We thank Reviewer EwJC for this comment. We have revised Sec 2.4 to provide provide a high level account, and split off a more precise definition to Appendix H.
>
> > It seems to me the big 'insight' is Turing completeness of the model, but this is (mostly) done in previous papers. Once this is established the remaining reductions are much less surprising.
>
> We respectfully disagree with the assessment that the big insight of this paper is the Turing-completeness of modern neural model families. The main focus of this work is on weighted languages, but the common definition of Turing-completeness does not consider weighting strings. Speaking at a high level, one of our main messages is that the uncomputability arises in the combination of Turing-completeness **and** the ability of assigning relatively high weights to long strings. Autoregressive model families which are based on Turing-complete models do not suffer from uncomputability results of this paper (Theorem 9): they do not have the ability to assign arbitrarily high scores to long strings, and are thus weak (see Lin et al. 2021).
>
> > I'm not sure the paper is a good fit for ICLR. It is mainly a complexity/computability paper where the models are related to ML.
>
> We believe ICLR is the **perfect** venue for this paper: our paper falls in the category of ‘theoretical issues in deep learning’ from the ICLR 2022 CfP. We ourselves are ML practitioners, and theoretical results presented in our submission have been guiding the model design of other ongoing projects of ours. We expect this is the case for the ongoing work of others, too.

---

> > ### Comment · Reviewer_EwJC · 2021-11-17
> > **RE:**
> >
> > I thank the authors for a thoughtful response.

---

### Decision · Program_Chairs · 2022-01-20

**Decision:**

Accept (Spotlight)

**Comment:**

This is a deep theoretical paper with results that I consider very interesting. I have *not* had time to check them myself, but I have background in these theoretical matters and the results seem reasonable to me - the hardness of even checking the quality of a solution is well known for partition functions (as well as hardness of any reasonable approximation), but the undecidability seems new - I assume it comes naturally and it is a very interesting result - I have seen similar decidability issues for #P: general probabilistic polynomial time Turing machines (it is unclear if a connection was sought here). Reviewers are all positive about the content, and authors have acknowledge some points for improvement.